# Mitochondrial electron transport chain, ceramide, and coenzyme Q are linked in a pathway that drives insulin resistance in skeletal muscle

Alexis Diaz-Vegas[1], Søren Madsen[1], Kristen C Cooke[1], Luke Carroll[1], Jasmine XY Khor[1,2], Nigel Turner[3], Xin Y Lim[2], Miro A Astore[4], Jonathan C Morris[5], Anthony S Don[2], Amanda Garfield[6], Simona Zarini[6], Karin A Zemski Berry[6], Andrew P Ryan[7], Bryan C Bergman[6], Joseph T Brozinick[7], David E James[1,2]*, James G Burchfield[1]*

[1]Charles Perkins Centre, School of life and Environmental Sciences, University of Sydney, Sydney, Australia; [2]Charles Perkins Centre and School of Medical Sciences, Faculty of Medicine and Health, University of Sydney, Sydney, Australia; [3]Cellular Bioenergetics Laboratory, Victor Chang Cardiac Research Institute, Sydney, Australia; [4]Center for Computational Biology and Center for Computational Mathematics, Flatiron Institute, New York, United States; [5]School of Chemistry, UNSW Sydney, Sydney, Australia; [6]Division of Endocrinology, Metabolism, and Diabetes, University of Colorado Anschutz Medical Campus, Aurora, United States; [7]Lilly Research Laboratories, Division of Eli Lilly and Company, Indianapolis, United States

*For correspondence:
david.james@sydney.edu.au (DEJ);
james.burchfield@sydney.edu.au (JGB)

**Abstract** Insulin resistance (IR) is a complex metabolic disorder that underlies several human diseases, including type 2 diabetes and cardiovascular disease. Despite extensive research, the precise mechanisms underlying IR development remain poorly understood. Previously we showed that deficiency of coenzyme Q (CoQ) is necessary and sufficient for IR in adipocytes and skeletal muscle (Fazakerley et al., 2018). Here, we provide new insights into the mechanistic connections between cellular alterations associated with IR, including increased ceramides, CoQ deficiency, mitochondrial dysfunction, and oxidative stress. We demonstrate that elevated levels of ceramide in the mitochondria of skeletal muscle cells result in CoQ depletion and loss of mitochondrial respiratory chain components, leading to mitochondrial dysfunction and IR. Further, decreasing mitochondrial ceramide levels in vitro and in animal models (mice, C57BL/6J) (under chow and high-fat diet) increased CoQ levels and was protective against IR. CoQ supplementation also rescued ceramide-associated IR. Examination of the mitochondrial proteome from human muscle biopsies revealed a strong correlation between the respirasome system and mitochondrial ceramide as key determinants of insulin sensitivity. Our findings highlight the mitochondrial ceramide–CoQ–respiratory chain nexus as a potential foundation of an IR pathway that may also play a critical role in other conditions associated with ceramide accumulation and mitochondrial dysfunction, such as heart failure, cancer, and aging. These insights may have important clinical implications for the development of novel therapeutic strategies for the treatment of IR and related metabolic disorders.

## eLife assessment

This **important** study highlights a potential connection between fatty acid intrusion into myocytes and increases in mitochondrial ceramide that cause deficits in coenzyme Q and consequent insulin

resistance. The authors primarily use the L6 myocyte model, which may not fully recapitulate in vivo conditions; however, the article shows **compelling** data in mice that substantially supports the L6 cell results. Overall, this study provides a strong framework for a **compelling** pathway of myocyte dysfunction and for continued efforts to test the **important** hypotheses that are presented.

## Introduction

Insulin is the primary hormone responsible for lowering blood glucose, in part, by stimulating glucose transport into muscle and adipose tissue. This is mediated by the phosphatidylinositol 3-kinase/Akt-dependent delivery of insulin-sensitive glucose transporters (GLUT4) to the plasma membrane (PM) (*Hill et al., 1999*; *Cong et al., 1997*). This process is defective in insulin resistance, a significant risk factor for cardiometabolic diseases such as type 2 diabetes (*James et al., 2021*), heart failure (*Riehle and Abel, 2016*), and some types of cancer (*Leitner et al., 2022*), and so defective GLUT4 translocation represents one of the hallmarks of insulin resistance.

The development of insulin resistance in skeletal muscle and adipocytes has been associated with multiple intracellular lesions, including mitochondrial coenzyme Q (CoQ) deficiency (*Fazakerley et al., 2018a*), accumulation of intracellular lipids such as ceramides (*Holland and Summers, 2008*), and increased mitochondrial reactive oxygen species (ROS) (*Anderson et al., 2009*; *Hoehn et al., 2009*; *Fazakerley et al., 2018b*). However, delineating the relative contribution of these lesions to whole-body insulin resistance and their interconnectivity remains a challenge.

CoQ (CoQ9 in rodents and CoQ10 in humans) is a mitochondrial cofactor and antioxidant synthesized and localized in the inner mitochondrial membrane (IMM). This cofactor is essential for mitochondrial respiration (*Hatefi et al., 1962*), fatty acid oxidation (*Frerman, 1988*), and nucleotide biosynthesis (*Jones, 1980*). We reported that mitochondrial, but not global, CoQ9 depletion is both necessary and sufficient to induce insulin resistance in vitro and in vivo (*Fazakerley et al., 2018a*), suggesting a causal role of CoQ9/10 depletion in insulin resistance. CoQ deficiency can result from primary mutation in the CoQ biosynthetic machinery (named complex Q) (*Montini et al., 2008*) or secondary from other cellular defects such as deletion of the oxidative phosphorylation system (OXPHOS) (*Kühl et al., 2017*; *Calvo et al., 2020*). Low levels of CoQ10 are associated with human metabolic disease, including diabetes (*Ates et al., 2013*; *El-ghoroury et al., 2009*), cardiovascular disease (*Zozina et al., 2018*), and aging (*de Barcelos and Haas, 2019*). Strikingly, many of these conditions are also associated with loss of OXPHOS and mitochondrial dysfunction (*Højlund et al., 2010*; *Diaz-Vegas et al., 2020*). However, it is unclear what causes these mitochondrial defects or if they are mechanistically linked.

Ceramides belong to the sphingolipid family, and high levels are strongly associated with insulin resistance (*Chaurasia and Summers, 2021*). While it has been proposed that ceramides cause insulin resistance by inhibition of PI3K/Akt signaling (*Summers et al., 1998*; *Schubert et al., 2000*; *Powell et al., 2003*), there is now considerable evidence that does not support this (*Fazakerley et al., 2018a*; *James et al., 2021*; *Kono and Barham, 1971*; *Hoehn et al., 2008*). Hence, ceramides may induce insulin resistance by a non-canonical mechanism. The rapid decline of mitochondrial oxidative phosphorylation in isolated mitochondria in the presence of N-acetylsphingosine (C2-ceramide), a synthetic ceramide analog that can penetrate cells, suggests that ceramides may be responsible for defective mitochondria (*Di Paola et al., 2000*). This effect seems to be ceramide-specific as neither diacylglycerides (DAGs) nor triacylglycerides (TAGs) affect mitochondrial respiration (*Perreault et al., 2018*). Recent evidence suggests a link between mitochondrial ceramides and insulin sensitivity, with the observation that reducing mitochondrial, but not global, ceramide in the liver protects against the development of diet-induced insulin resistance and obesity (*Perreault et al., 2018*; *Hammerschmidt et al., 2019*). Consistent with this, mitochondrial ceramide levels are more strongly associated with insulin resistance than with whole-tissue ceramide in human skeletal muscle (*Perreault et al., 2018*). Despite this association, no direct evidence exists linking mitochondrial ceramides with insulin sensitivity in skeletal muscle.

Here, we describe the linkage between mitochondrial ceramide, CoQ, OXPHOS, and ROS in the etiology of insulin resistance. We show that a strong inverse relationship between mitochondrial CoQ and ceramide levels is intimately linked to the control of cellular insulin sensitivity. For example, increasing mitochondrial ceramide using either chemical or genetic tools, decreased mitochondrial

CoQ levels, and induced insulin resistance. Conversely, genetic or pharmacological manipulations that lowered mitochondrial ceramide levels increased CoQ levels and protected against insulin resistance. Increased mitochondrial ceramides also led to a reduction in several OXPHOS components, hindering mitochondrial respiration and elevating mitochondrial ROS in vitro. This was further supported in human skeletal muscle, where we observed a strong association between insulin sensitivity, abundance of OXPHOS, and mitochondrial ceramides. We propose that increased mitochondrial ceramides cause a depletion in various OXPHOS components, leading to mitochondrial malfunction and deficiency in CoQ, resulting in increased ROS and insulin resistance. This provides a significant advance in our understanding of how ceramide causes mitochondrial dysfunction and insulin resistance in mammals.

## Results
### Palmitate induces insulin resistance by increasing ceramides and lowering CoQ9 levels in L6 myotubes

Lipotoxicity plays a major role in insulin resistance and cardiometabolic disease (*Ertunc and Hotamisligil, 2016*). Excess lipids accumulate in insulin target tissues, such as muscle, impairing insulin-stimulated GLUT4 translocation, as well as other metabolic actions of insulin. For this reason, several in vitro models have been employed involving incubation of insulin-sensitive cell types with lipids such as palmitate to mimic lipotoxicity in vivo (*Hoehn et al., 2009*). In this study, we have used cell surface GLUT4-HA abundance as the main readout of insulin response. As shown (*Figure 1A*), incubation of L6 myotubes with palmitate (150 µM for 16 hr) reduced the insulin-stimulated translocation of GLUT4 to the cell surface by ~30%, consistent with impaired insulin action. Despite this marked defect in GLUT4 translocation, we did not observe any defect in proximal insulin signaling as measured by phosphorylation of Akt or TBC1D4 (*Figure 1C and D* – 100 nM insulin; 20 min). Previous studies linking ceramides to defective insulin signaling have utilized the short-chain ceramide analog (C2-ceramide) (*Summers et al., 1998*; *Schubert et al., 2000*; *Powell et al., 2003*). Intriguingly, we were able to replicate that C2-ceramide inhibited both GLUT4 translocation and Akt phosphorylation in L6 myocytes (*Figure 1B–D*). One possibility is that palmitate induces insulin resistance in L6 myotubes via a ceramide-independent pathway. However, this is unlikely as palmitate-induced insulin resistance was prevented by the ceramide biosynthesis inhibitor myriocin (*Figure 1A*) and we observed a specific increase in C16-ceramide levels in L6 cells following incubation with palmitate, which was also prevented by myriocin (*Figure 1E and F*, *Figure 1—figure supplement 1*). Based on these data, we surmise that C2-ceramide does not faithfully recapitulate physiological insulin resistance, in contrast to that seen with incubation with palmitate.

We previously demonstrated that insulin resistance was associated with CoQ depletion in muscle from high-fat diet (HFD)-fed mice (*Fazakerley et al., 2018a*). To test if CoQ supplementation reversed palmitate-induced insulin resistance, L6 myotubes were co-treated with palmitate plus CoQ9. Addition of CoQ9 had no effect on control cells but overcame insulin resistance in palmitate-treated cells (*Figure 1A*). Notably, the protective effect of CoQ9 appears to be downstream of ceramide accumulation, as it had no impact on palmitate-induced ceramide accumulation (*Figure 1E and F*). Strikingly, both myriocin and CoQ9 reversed insulin resistance, suggesting that there might be an interaction between ceramides and CoQ in the induction of insulin resistance with palmitate in these cells. Moreover, we have previously shown that mitochondrial CoQ is a key determinant of insulin resistance (*Fazakerley et al., 2018a*), suggesting that ceramides and CoQ may interact in mitochondria. To explore this link, we next examined the effect of palmitate on mitochondrial CoQ levels. As shown (*Figure 1G*), palmitate lowered mitochondrial CoQ9 abundance by ~40%, and this was prevented with myriocin. To test whether CoQ depletion is downstream of ceramide accumulation, we exposed GLUT4-HA-L6 myotubes to 4-nitrobenzoic acid (4NB) to competitively inhibit 4-hydroxybenzoate:prolyprenyl transferase (Coq2), a limiting step in CoQ9 synthesis (*Forsman et al., 2010*). 4NB (2.5 mM for 16 hr) decreased mitochondrial CoQ9 to a similar extent as observed in palmitate-treated myocytes (*Figure 1I*) and generated insulin resistance in GLUT4-HA-L6 myotubes (*Figure 1H*). Notably, 4NB-mediated insulin resistance was prevented by provision of CoQ9, as described previously (*Fazakerley et al., 2018a*). Interestingly, total ceramide abundance was increased in 4NB-treated cells, albeit to a lesser extent than observed with palmitate, without affecting other lipid species (*Figure 1J and K*, *Figure 1—figure supplement 1*). One possibility is that CoQ directly

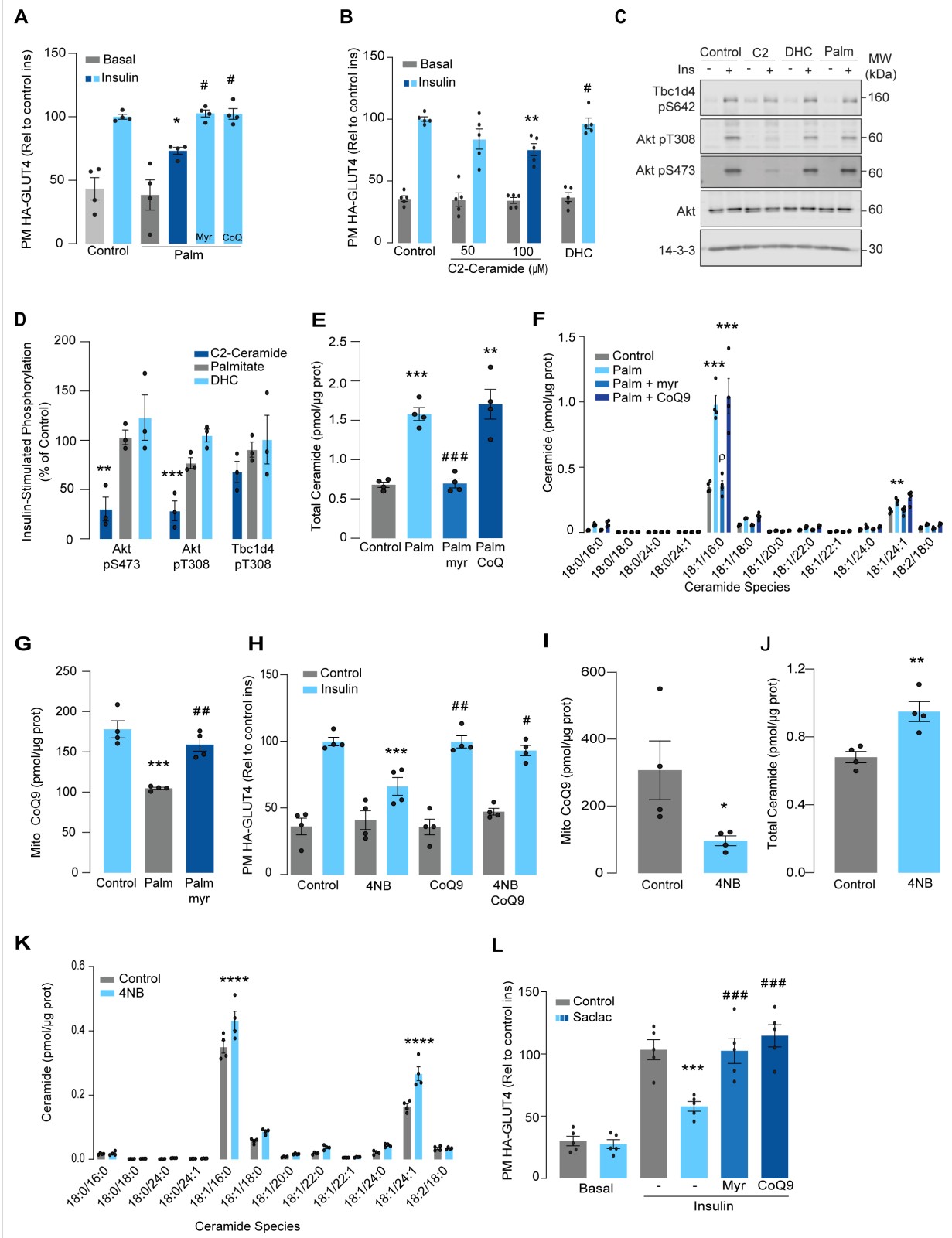

**Figure 1.** Palmitate increases ceramides, decreases coenzyme Q (CoQ), and induces insulin resistance in L6 myotubes. (**A**) Insulin-induced GLUT4 translocation in L6-HA-GLUT4 myotubes exposed to palmitate (150 μM for 16 hr, Palm) or BSA (Control) in the presence of DMSO (Control), myriocin (10 μM for 16 hr), or CoQ9 (10 μM for 16 hr). Plasma membrane GLUT4 (PM-GLUT4) abundance was normalized to insulin-treated control cells. N = 4 biological replicates, mean ± SEM. *p<0.05 vs Control ins, # p<0.5 vs Palm ins. (**B**) Insulin-induced GLUT4 translocation in L6-HA-GLUT4 myotubes

*Figure 1 continued on next page*

*Figure 1 continued*

exposed to C2-ceramide, dihydroceramide (100 µM, DHC) or DMSO (Control) for 2 hr. Plasma membrane GLUT4 (PM-GLUT4) abundance was normalized to insulin-treated control cells. N = 5 biological replicates, mean ± SEM. **p<0.01 vs Control ins, # p<0.5 vs 100 µM C2-ceramide ins. (**C, D**) L6-HA-GLUT4 myotubes were serum-starved after BSA (Control for 16 hr), palmitate (Palm for 16 hr), C2-ceramide (100 µM for 2 hr, C2), or DHC (100 µM for 2 hr) treatment and acute insulin (Ins) was added where indicated. Phosphorylation status of indicated sites was assessed by immunoblot (**C**). Immunoblots were quantified by densitometry and normalized to insulin-treated control cells (indicated by dotted line). N = 3 biological replicates, mean ± SEM. **p<0.01, ***p<0.001. (**E, F**) Endogenous ceramides levels in L6-HA-GLUT4 myotubes treated for 16 hr with BSA (control), palmitate (150 µM, Palm), myriocin (10 µM for 16 hr), or CoQ (10 µM for 16 hr) as indicated in the graph. Total (**E**) and specific (**F**) ceramide species were quantified. N = 4 biological replicates, mean ± SEM. **p<0.01, ****p<0.001 vs Control, ###p<0.001 vs Palm, $\rho$ p<0.01 vs Palm. (**G**) CoQ9 level in mitochondrial fraction obtained from L6-HA-GLUT4 myotubes. N = 4 biological replicates, mean ± SEM. ***p<0.001 vs Control, ##p<0.01 vs Palm. (**H**) Insulin-induced GLUT4 translocation in L6-HA-GLUT4 myotubes exposed to 4-NB (2.5 mM for 16 hr) or DMSO (Control) in the presence of CoQ9 (10 µM for 16 hr). Plasma membrane GLUT4 (PM-GLUT4) abundance was normalized to insulin-treated control cells. N = 4 biological replicates, mean ± SEM. ***p<0.001 vs Control ins, ##p<0.01, ###p<0.001 vs 4NB. (**I**) CoQ9 level in mitochondrial fraction obtained from L6-HA-GLUT4 myotubes exposed to DMSO (Control) or 4NB for 16 hr. N = 4 biological replicates, mean ± SEM. *p<0.05 (**J, K**) Total (**J**) and specific (**K**) ceramide species quantified in L6-HA-GLUT4 myotubes treated for 16 hr with DMSO (control) or 4NB (2.5 mM for 16 hr). N = 4 biological replicates, mean ± SEM. **p<0.01, ****p<0.0001. (**L**) Insulin-induced GLUT4 translocation in L6-HA-GLUT4 myotubes exposed to Saclac (10 µM for 24 hr) or EtOH (Control) in the presence of DMSO (control), myriocin (10 µM for 16 hr), or CoQ9 (10 µM for 16 hr). Plasma membrane GLUT4 (PM-GLUT4) abundance was normalized to insulin-treated control cells. N = 5 biological replicates, mean ± SEM. ***p<0.001 vs Control Ins, ### p<0.001 vs Saclac Ins.

The online version of this article includes the following figure supplement(s) for figure 1:

**Figure supplement 1.** Inhibiting ceramidases is sufficient to decrease CoQ levels, leading to insulin resistance.

**Figure supplement 2.** Inhibtion of ceramidases induces CoQ depletion in human cell line.

controls ceramide turnover (*Fernández-Ayala et al., 2000*). An alternate possibility is that CoQ inside mitochondria is necessary for fatty acid oxidation (*Frerman, 1988*) and CoQ depletion triggers lipid overload in the cytoplasm promoting ceramide production (*Koves et al., 2008*). In fact, increased fatty acid oxidation is protective against insulin resistance in several model organisms (*Bruce et al., 2009*; *Sebastián et al., 2007*; *Perdomo et al., 2004*). Future studies are required to determine how CoQ depletion promotes Cer accumulation. Regardless, these data indicate that ceramide and CoQ have a central role in regulating cellular insulin sensitivity.

Since palmitate treatment can have a number of effects beyond ceramides, we next attempted to increase intracellular ceramides by inhibiting the ceramide degradation pathway. We exposed L6 myotubes to different concentrations of Saclac, an inhibitor of acid ceramidase (*Kao et al., 2019*), for 24 hr. Saclac increases ceramides in L6 cells in a dose-dependent fashion, with the largest effect on C16:0 ceramides (*Figure 1—figure supplement 1F*). Interestingly, Saclac also promoted accumulation of DAGs, sphingosine-1 phosphate (S1P), and sphingosine (SPH), demonstrating the tight interaction between these lipid species (*Figure 1—figure supplement 1B, C, and E*). Consistent with a role of ceramide in insulin sensitivity, Saclac (10 µM for 24 hr) reduced insulin-stimulated GLUT4 translocation by 40% (*Figure 1L*, vs Control; p<0.001), and this was prevented by myriocin or CoQ9 supplementation (*Figure 1L*). Notably, no detectable defects in Akt phosphorylation were observed (*Figure 1—figure supplement 1G and H*).

To explore if ceramides promote CoQ depletion beyond skeletal muscle, human cervical cancer cells (HeLa) were exposed to Saclac, as described previously (2 µM for 24 hr) (*Fisher-Wellman et al., 2021*). Consistent with our observation in L6 myotubes, Saclac increased total ceramide levels (approximately sixfold over basal) (*Figure 1—figure supplement 1F*) and lowered CoQ levels inside mitochondria (*Figure 2—figure supplement 1C*). Of interest, myriocin prevented Saclac-induced-CoQ depletion, demonstrating that there is a similar interaction between ceramide and CoQ levels in this human cell line as observed in L6 cells (*Figure 2—figure supplement 1D*). Moreover, this was relatively specific to CoQ as we did not observe any change in mitochondrial mass with Saclac (*Figure 2—figure supplement 1E–G*), cell viability (*Figure 2—figure supplement 1H*), or DAG abundance (*Figure 2—figure supplement 1I*). Regardless, these data indicate that there is a strong association between ceramide and CoQ and that this has a central role in regulating cellular insulin sensitivity.

## Mitochondrial ceramide promotes insulin resistance by lowering CoQ levels

Although mitochondrial ceramides have been linked with insulin resistance in human skeletal muscle (*Perreault et al., 2018*), to date, there is no direct evidence linking mitochondrial ceramides with insulin sensitivity. We wanted to determine whether ceramide accumulation specifically in mitochondria is associated with altered CoQ levels and insulin resistance. To achieve this, we employed doxycycline-Tet-On-inducible (*Das et al., 2016*) overexpression of a mitochondrial-targeted sphingomyelin phosphodiesterase 5 (mtSMPD5) in GLUT4-HA-L6 cells (GLUT4-HA-L6-mtSMPD5) (*Figure 2A*). SMPD5 is a murine mitochondria-associated enzyme (*Wu et al., 2010*) that hydrolyzes sphingomyelin to produce ceramides (*Bienias et al., 2016*). Thus, overexpressing mtSMPD5 should specifically increase ceramides within mitochondria and avoid potential nonspecific effects associated with small-molecule inhibitors. As expected, doxycycline induced mitochondrial expression of mtSMPD5, as demonstrated by enrichment of SMPD5 in mitochondria isolated from L6 cells (*Figure 2B*), and this was associated with increased total mitochondrial ceramides to the same extent as observed with palmitate treatment (*Figure 2C*), with the largest increase in C16-ceramide (*Figure 2D*). Importantly, mtSMPD5 overexpression did not affect ceramide abundance in the whole-cell lysate nor other lipid species inside mitochondria such as cardiolipin, cholesterol, and DAGs (*Figure 3—figure supplement 1A and D–J*). Intriguingly, mtSMD5 did not affect sphingomyelin levels in mitochondria (*Figure 3—figure supplement 1G*), consistent with exchange between mitochondrial and extra-mitochondrial sphingomyelin pools to compensate for the degradation induced by SMPD5 overexpression (*Feng et al., 2018*). Consistent with our hypothesis, mtSMPD5 was sufficient to promote insulin resistance in response to submaximal and maximal insulin doses (*Figure 2E*). Furthermore, mtSMPD5 overexpression promoted insulin resistance without affecting Akt phosphorylation (*Figure 2F–H*), and no differences in total GLUT4 levels were observed across the treatments (*Figure 2F*, *Figure 3—figure supplement 1B*). We next explored if mitochondrial ceramide-induced insulin resistance was mediated by lowering CoQ within mitochondria. In line with our previous results, mitochondrial CoQ levels were depleted in both palmitate-treated and mtSMPD5-overexpressing cells without any additive effects. This suggests that these strategies to increase ceramides share a common mechanism for inducing CoQ depletion in L6 myotubes (*Figure 2I*). Importantly, CoQ9 supplementation prevented both palmitate- and mtSMPD5-induced insulin resistance (*Figure 2J*), suggesting that CoQ depletion is an essential mediator of insulin resistance.

## Mitochondrial ceramides are necessary for palmitate-induced CoQ depletion and insulin resistance

Given that increased mitochondrial ceramides are sufficient to induce CoQ depletion and insulin resistance, we next asked whether increased mitochondrial ceramides are necessary to drive these phenotypes. Using the doxycycline-Tet-On-inducible system (*Das et al., 2016*), we overexpressed a mitochondrial-targeted acid ceramidase 1 (mtASAH1) in GLUT4-HA-L6 cells (GLUT4-HA-L6-mtASAH1) (*Figure 3A*). ASAH1 degrades ceramides to fatty acid and sphingosine (*Li et al., 1999*). Hence, mitochondrial overexpression of ASAH1 was expected to selectively lower ceramides inside mitochondria. Doxycycline increased the abundance of mtASAH1 in the mitochondrial fraction, demonstrating the correct localization of this construct (*Figure 3B*). Furthermore, mtASAH1 induction prevented palmitate-induced mitochondrial ceramide accumulation (total levels and 18:1\16:0 ceramides) (*Figure 3C*), indicating that the enzyme was functioning as expected. Similar to our observations with mtSMD5 overexpression, mtASAH1 did not alter ceramide abundance in the whole-cell lysate or mitochondrial sphingomyelin levels (*Figure 3—figure supplement 1A and F*). Notably, mtASAH1 overexpression protected cells from palmitate-induced insulin resistance without affecting basal insulin sensitivity (*Figure 3E*). Similar results were observed using insulin-induced glycogen synthesis as an orthologous technique for GLUT4 translocation. These results provide additional evidence highlighting the role of dysfunctional mitochondria in muscle cell glucose metabolism (*Figure 3—figure supplement 1K*). Importantly, mtASAH1 overexpression did not rescue insulin sensitivity in cells depleted of CoQ (2.5 mM 4NB for 24 hr), supporting the notion that mitochondrial ceramides are upstream of CoQ (*Figure 3E*). Neither palmitate nor mtASAH1 overexpression attenuated insulin-dependent Akt phosphorylation (*Figure 3F–H*) nor total GLUT4 abundance (*Figure 3—figure supplement 1B*). Finally, mtASAH1 overexpression increased CoQ levels. In both control and mtASAH1 cells, palmitate

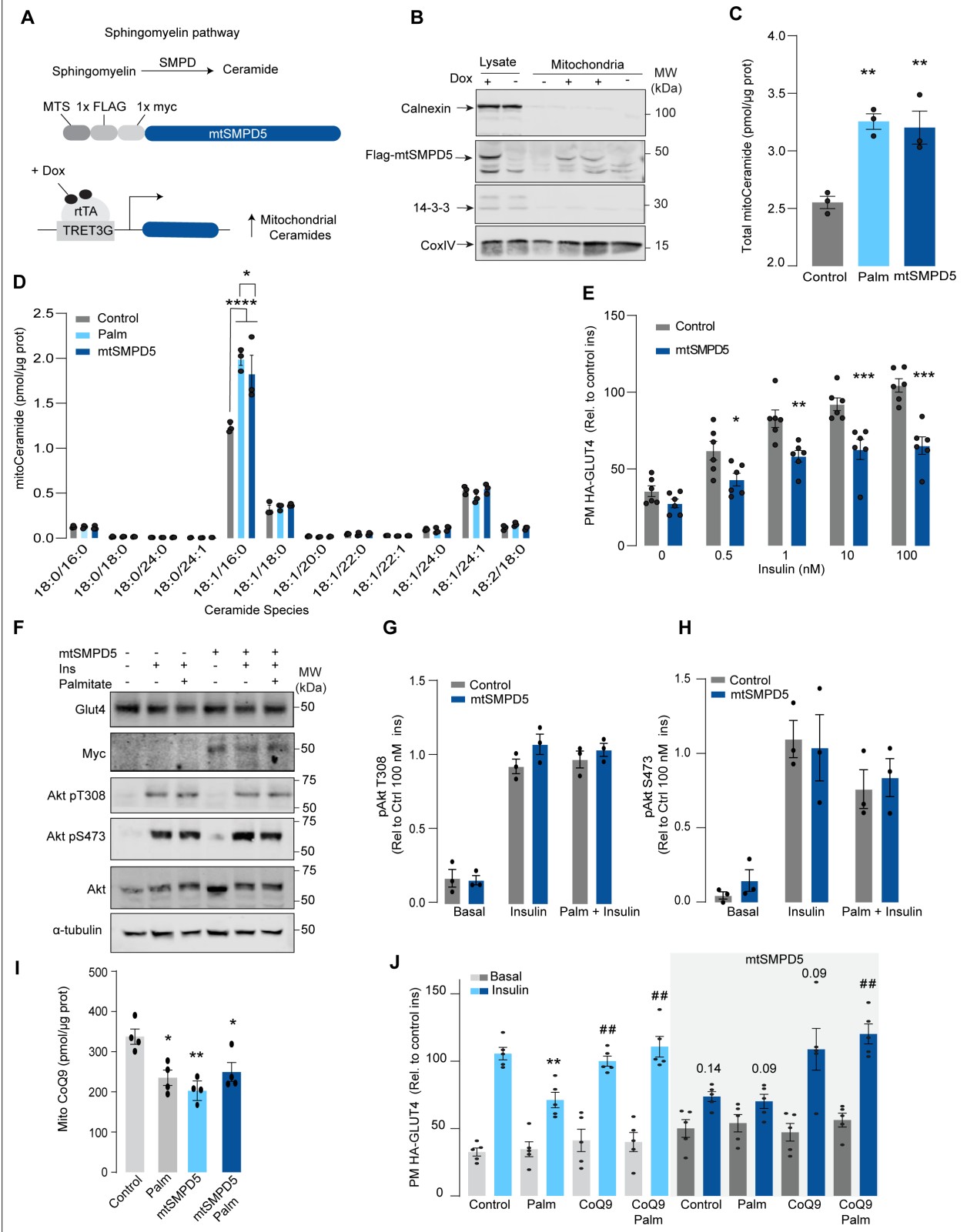

**Figure 2.** Mitochondrial overexpression of SMPD5 induces insulin resistance by lowering CoQ9 in L6 myotubes. (**A**) Schematic representation of doxycycline-inducible overexpression of mitochondrial-targeted sphingomyelinase 5 (SMPD5). L6-HA-GLUT4 myotubes were exposed to 1 μg/mL of doxycycline from day 3 to day 6 of differentiation. Experiments were performed on day 7 of differentiation. (**B**) Determination of SMPD5 expression in mitochondrial fraction obtained from L6-HA-GLUT4. Doxycycline was added where indicated. (**C, D**) Levels of endogenous ceramides in mitochondrial

*Figure 2 continued on next page*

Figure 2 continued

fraction from L6-HA-GLUT4 myotubes treated with BSA (Control), palmitate (150 µM for 16 hr, Palm), or doxycycline. Total (**C**) and specific (**D**) ceramide species were quantified. N = 3 biological replicates, mean ± SEM. *p<0.05, **p<0.01, ****p<0.0001 vs Control. (**E**) Insulin-induced GLUT4 translocation in L6-HA-GLUT4 myotubes exposed to doxycycline (1 µg/mL for 3 d). Plasma membrane GLUT4 (PM-GLUT4) abundance was normalized to 100 nM insulin-treated control cells. N = 6 biological replicates, mean ± SEM. *p<0.05, **p<0.01, ***p<0.001 vs Control ins. (**F–H**) L6-HA-GLUT4 myotubes were serum-starved after BSA (Control), palmitate (150 µM for 16 hr, Palm), or doxycycline (1 µg/mL for 3 d) treatment and acute insulin (Ins) was added where indicated. Phosphorylation status of indicated sites was assessed by immunoblot. Immunoblots were quantified by densitometry and normalized to insulin-treated control cells (indicated by dotted line). N = 3 biological replicates, mean ± SEM. * p<0.05, *** p<0.001 vs Basal. (**I**) CoQ9 level in mitochondrial fraction obtained from L6-HA-GLUT4 myotubes exposed to doxycycline (1 µg/mL for 3 d). N = 4 biological replicates, mean ± SEM. **p<0.001. (**J**) Insulin-induced GLUT4 translocation in L6-HA-GLUT4 myotubes exposed to doxycycline (1 µg/mL for 3 d). Control or doxycycline-treated cells were exposed to BSA (Control), palmitate (150 µM for 16 hr, Palm), or CoQ9 (10 µM for 16 hr). Plasma membrane GLUT4 (PM-GLUT4) abundance was normalized to insulin-treated control cells. N = 5 biological replicates, mean ± SEM. **p<0.01 vs Control ins, ## p<0.01 vs Palm ins.

The online version of this article includes the following figure supplement(s) for figure 2:

**Figure supplement 1.** The overexpression of mtSMPD5 selectively increases mitochondrial ceramides.

induced a depletion of CoQ; however, the levels in palmitate-treated mtASAH1 cells remained similar to control untreated cells (**Figure 3I**). This suggests that the absolute concentration of CoQ is crucial for insulin sensitivity, rather than the relative depletion compared to basal conditions, thus supporting the causal role of mitochondrial ceramide accumulation in reducing CoQ levels in insulin resistance.

In order to demonstrate the connection between ceramide and CoQ in vivo, we examined whether a reduction in ceramides in mouse skeletal muscle, using a ceramide synthase 1 (CerS1) inhibitor, would alter mitochondrial CoQ levels. Treatment of adult mice with the CerS1 inhibitor P053 for 6 wk selectively lowered muscle ceramides without affecting other lipid species (**Figure 3J**, **Figure 3— figure supplement 2**). Notably, CerS1 inhibition increased CoQ in mitochondrial fractions isolated from skeletal muscle (**Figure 3K**), and a similar effect was observed in mice exposed to an HFD for 5 wk (**Figure 3—figure supplement 1H and I**). These animals exhibited an improvement in mitochondrial function and reduced muscle triglycerides and adiposity upon HFD (further phenotypic and metabolic characterization of these animals can be found in **Turner et al., 2018**), demonstrating the existence of the ceramide/CoQ relationship in muscle in vivo.

## Mitochondrial ceramides induce depletion of the electron transport chain

We have established that both increased mitochondrial ceramides and a loss of mitochondrial CoQ are necessary for the induction of insulin resistance. As such, these changes are likely to induce other mitochondrial defects. To gain insight into how increased mitochondrial ceramides drive changes in mitochondrial function, we performed MS-based proteomics on L6 cells overexpressing mtSMPD5.

mtSMPD5-L6 myotubes were treated with doxycycline for various time points (2, 8, 24, 48, and 72 hr) and positive induction was observed after 24 hr of treatment (**Figure 4—figure supplement 1A**). Subsequently, control, 24, and 72 hr time points were selected for further studies. Mitochondria were purified via gradient separation and analyzed using liquid chromatography-tandem mass spectrometry (LC-MS/MS) in data-independent acquisition (DIA) mode (**Figure 4A**). Across control and mtSMPD5 cells, we quantified 2501 proteins where 555 were annotated as mitochondrial proteins (MitoCarta 3.0 and UniProt localization) (**Rath et al., 2021**).

Analysis of the proteome revealed that 9 and 19% of mitochondrially annotated proteins were significantly changed at 24 and 72 hr, respectively (adj. p<0.05, absolute log2 FC > 0.4), indicative of a temporal progression of changes following induction of mSMPD5 expression (**Figure 4B and C**). Sixty proteins were decreased at 72 hr, and of these 47% were functionally annotated as components of oxidative phosphorylation (OXPHOS; rank 1, 28/135 proteins). Within OXPHOS, we observed a significant depletion of the electron transport chain components (ETC). The ETC is composed of several complexes (complexes I–V, CI–CV). In this dataset, CI (14/15 decreased), CIII (5/6 decreased), and CIV (13/13 decreased) but not CII (0/3 decreased) or CV (3/15 decreased) were depleted after mtSMPD5 overexpression (**Figure 4D**). Despite the bulk downregulation of CI, III, and IV, the assembly machinery associated with each complex was either upregulated or unchanged after mtSMPD5 overexpression (**Figure 4J**), suggesting that mitochondrial ceramides somehow alter ETC stability. Intriguingly, neither CII nor CV were affected by mtSMPD5, suggesting that ceramides preferentially affect those

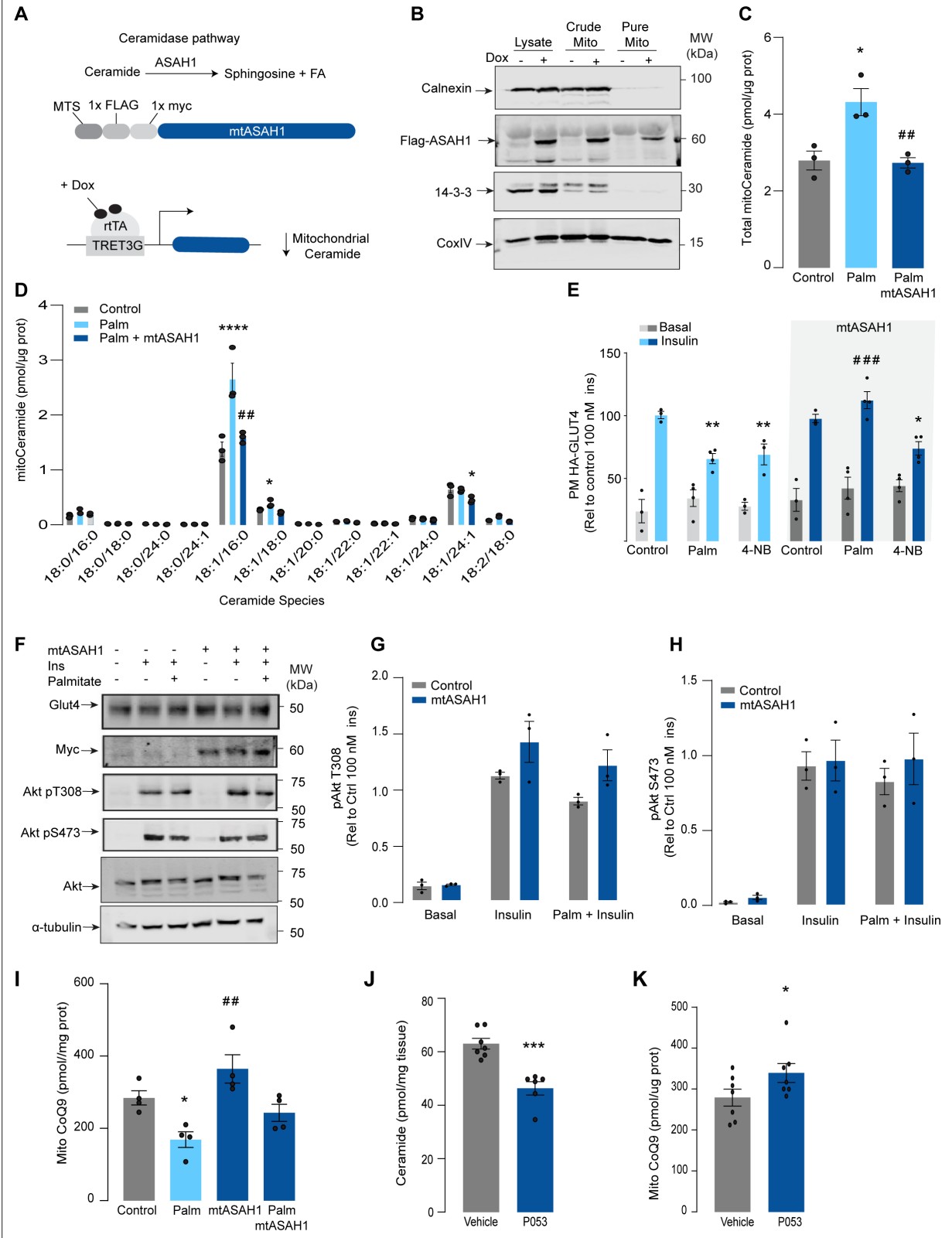

**Figure 3.** Mitochondrial overexpression of ASAH1 protects against insulin resistance and increases coenzyme Q (CoQ) levels in L6 myotubes. (**A**) Schematic representation of doxycycline-inducible overexpression of mitochondrial-targeted acid ceramidase 1 (ASAH1). L6-HA-GLUT4 myotubes were exposed to 1 µg/mL of doxycycline from day 3 to day 6 of differentiation. Experiments were performed on day 7 of differentiation. (**B**) Determination of ASAH1 expression in mitochondrial fraction obtained from L6-HA-GLUT4. Doxycycline was added where indicated. (**C, D**) Endogenous ceramides

*Figure 3 continued on next page*

*Figure 3 continued*

levels in mitochondrial fraction from L6-HA-GLUT4 myotubes treated with BSA (Control), palmitate (150 µM for 16 hr, Palm), or doxycycline. Total (**C**) and specific (**D**) ceramide species were quantified. N = 3 biological replicates, mean ± SEM. *p<0.05 vs Control, ##p<0.01 vs Palm. (**E**) Insulin-induced GLUT4 translocation in L6-HA-GLUT4 myotubes exposed to doxycycline (1 µg/mL for 3 d). Control or doxycycline-treated cells were exposed to BSA (control), palmitate (150 µM for 16 hr, Palm), or 4NB (2.5 mM for 16 hr). Plasma membrane GLUT4 (PM-GLUT4) abundance was normalized to insulin-treated control cells. N = 6 biological replicates, mean ± SEM. **p<0.01 vs Control ins, ### p<0.001 vs Palm ins. (**F–H**) L6-HA-GLUT4 myotubes were serum-starved after BSA (Control), palmitate (150 µM for 16 hr, Ppalm), or doxycycline (1 µg/mL for 3 d) treatment and acute insulin (Ins) was added where indicated. Phosphorylation status of indicated sites was assessed by immunoblot. Immunoblots were quantified by densitometry and normalized to insulin-treated control cells (indicated by dotted line). N = 3 biological replicates, mean ± SEM. ***p<0.001 vs Basal, # p<0.05 vs Control ins. (**I**) CoQ9 level in mitochondrial fraction obtained from L6-HA-GLUT4 myotubes exposed to doxycycline (1 µg/mL for 3 d). Control or doxycycline treated cells were exposed to BSA (Control) or palmitate (150 µM for 16 hr, Palm) N = 4 biological replicates, mean ± SEM. *p<0.05 vs Control, ## p<0.01 vs Palm. (**J**) Levels of total ceramides in skeletal muscle of mice fed chow with vehicle or 5 mg/kg P053 for 6 wk. N = 7 mice per group, mean ± SEM. ***p<0.001. (**K**) Levels of CoQ in mitochondrial fraction isolated from skeletal muscle of mice fed chow with vehicle or 5 mg/kg P053 for 6 wk. N = 7 mice per group, mean ± SEM. *p<0.05.

The online version of this article includes the following figure supplement(s) for figure 3:

**Figure supplement 1.** The overexpression of mtASAH1 decreases DAGs in palmitate-treated cells.

**Figure supplement 2.** Lipidomic analysis of mice exposed to the CerS1 inhibitor P053.

ETC complexes that are part of structures known as supercomplexes (SCs) (*Letts and Sazanov, 2017*). Importantly, as part of CoQ is found in SCs binding CI (*Figure 4F*), we mapped the levels of individual subunits of CI onto the recently solved structure of bovine CI (*Zhu et al., 2016*). This revealed the loss of subunits around the N-module (Ndufs1, Ndufs4, Ndufs6, Ndufv2, and Ndufv3) and Q-module (Ndufa5 and Ndufs2) in CI (*Figure 4H*). Importantly, CI downregulation was not associated with reduction in gene expression as shown in *Figure 4—figure supplement 1J*. The N-module is essential for NADH oxidation, and the Q-module is where CoQ binds CI. Hence, loss of the Q-module might trigger a stoichiometric depletion of CoQ upon ceramide accumulation.

Of note, we observed a heterogeneous response of the mitochondrial proteome after mtSMPD5 overexpression. For instance, proteins associated with glucose oxidation and mitochondrial translation/transcription did not change after mtSMPD5 induction (*Figure 4—figure supplement 1D, F, and G*), proteins involved in fatty acid oxidation and OXPHOS were consistently downregulated after 24 hr treatment (*Figure 4B–D*, *Figure 4—figure supplement 1E*), proteins related with the mitochondrial import machinery were consistently upregulated (*Figure 4—figure supplement 1C*), and proteins associated with CoQ production were transiently downregulated after 24 h induction (*Figure 4E*).

## Mitochondrial ceramides impair mitochondrial function

Based on the ceramide-dependent depletion of ETC members, we hypothesized that mitochondrial ceramides would impair mitochondrial function. To test this, we evaluated several aspects of mitochondrial function upon mtSMPD5 overexpression. Mitochondrial respiration is broadly considered to be the best measure for describing mitochondrial activity (*Diaz-Vegas et al., 2020*; *Brand and Nicholls, 2011*). Respiration was assessed in intact mtSMPD5-L6 myotubes treated with CoQ9 by Seahorse extracellular flux analysis. mtSMPD5 overexpression decreased basal and ATP-linked mitochondrial respiration (*Figure 5A–C*), as well as maximal, proton-leak and non-mitochondrial respiration (*Figure 5A, D, E, and F*), suggesting that mitochondrial ceramides induce a generalized attenuation in mitochondrial function. Notably, we did not observe evidence of energy deficiency in our model (data not shown). Interestingly, CoQ9 supplementation partially recovered basal and ATP-linked mitochondrial respiration, suggesting that part of the mitochondrial defects are induced by CoQ9 depletion. The attenuation in mitochondrial respiration is consistent with a depletion of the ETC subunits observed in our proteomic dataset (*Figure 4*). Since mitochondrial respiratory activity is limited by several factors, including nutrient supply, bioenergetic demands, among others (*Krycer et al., 2020*), we tested whether the ETC generally or a specific respiratory complex was affected by mtSMPD5 overexpression. We measured the activity of the respiratory chain by providing substrates for each respiratory chain complex to permeabilized cells and analyzed oxygen consumption. mtSMPD5-overexpressing cells exhibited attenuated mitochondrial respiration irrespective of the substrate provided (*Figure 5G–L*), supporting the notion that mitochondrial ceramides induce a generalized defect in mitochondrial respiration. In line with defective mitochondrial function, cells

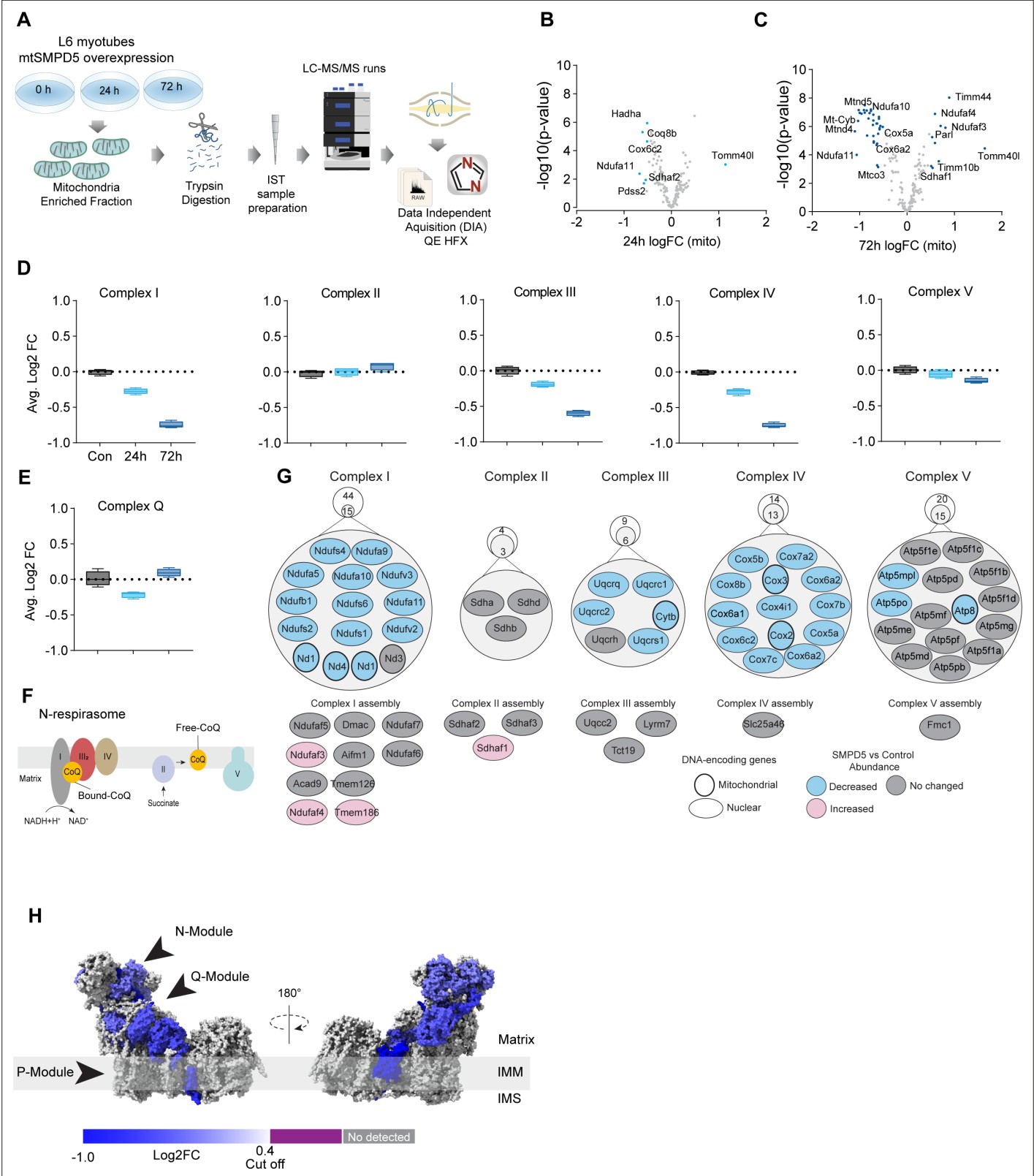

**Figure 4.** Mitochondrial ceramides induce a selective depletion of supercomplexes-associated proteins in L6 myotubes. (**A**) Workflow schematics. (**B, C**) Pairwise comparisons of mitochondrial proteome between all four groups. Cutoff for -log10 adjusted p-value (-log10(p-value)) was set at 2 and Log2(FC) at 0.5 (blue). (**D**) Quantification of the OXPHOS protein complexes generated by the summed abundance of all subunits within a given complex. (**E**) Quantification of the complex Q (CoQ) protein complexes generated by the summed abundance of all subunits within the complex. (**F**) Schematics

*Figure 4 continued on next page*

*Figure 4 continued*

of CoQ distribution between CI-binding and free CoQ (*Hernansanz-Agustín and Enríquez, 2021*). (**G**) Schematics of OXPHOS subunits (top) and assembly factors (bottom) significantly upregulated (light red), downregulated (blue), and no change (gray) after 72 hr of mtSMPD5 overexpression. (**H**) Subunit levels for proteins after mtSMPD5 overexpression mapped to the complex I structure (*Zhu et al., 2016*). The colors were calculated with an in-house Python script, and the resultant model was rendered using ChimeraX. Gray, not detected; purple, below cutoff (Log2FC = 0.4).

The online version of this article includes the following figure supplement(s) for figure 4:

**Figure supplement 1.** Proteomic analysis of mitochondrial fractions obtained rom L6 myotubes overexpressing mtSMPD5.

with mtSMPD5 overexpression also exhibited increased oxidative stress (*Figure 5M*) measured by the redox-sensitive dye MitoSOX. Interestingly, no difference in mitochondrial membrane potential was observed across conditions (*Figure 5N*). Collectively, these data suggest that increased mitochondrial ceramides cause a loss of mitochondrial respiratory capacity and an increase in ROS production as a result of ETC depletion in L6 myotubes.

## Association of mitochondrial proteome with insulin sensitivity and mitochondrial ceramides in human muscle

To further characterize the effect of mitochondrial ceramides on ETC abundance in a more physiological context, we performed a cross-sectional study assessing the mitochondrial lipid profile and protein abundance in muscle biopsies obtained from four groups of people (athletes, lean, obese, and type 2 diabetics [T2D]). The demographic information and detailed lipidomic analysis of these individuals were reported previously (*Perreault et al., 2018*, *Figure 5—figure supplement 1A*). In line with our in vitro data, long-tail ceramides (C18:0) in the mitochondria/endoplasmic reticulum (ER)-enriched fraction, but not whole tissue, were inversely correlated with muscle insulin sensitivity (*Perreault et al., 2018*). To expand this observation, we employed proteomics analysis of the mitochondrial/ER fraction from the same subjects (*Figure 6A*). A total of 2058 unique protein groups were quantified in at least one sample, where 571 were annotated as mitochondrial associated proteins (Human MitoCarta 3.0) (*Rath et al., 2021*). After filtering (proteins in >50% of samples within each group), 492 mitochondrial proteins were reliably quantified across 67 samples (*Figure 6B*). We noted that the mitochondrial fraction from athletes were enriched for mitochondrial proteins, and this could be corrected by global median normalization (*Figure 5—figure supplement 1D*). Pairwise comparison of the mitochondrial proteome between all groups revealed differences between groups, although relatively small in effect size (*Figure 6C and D*). For instance, 16% of all mitochondrial proteins were significantly different between T2D and athletes (*Figure 6C*); however, 56% (45/80) of these proteins were changed by less than 1.5-fold. This trend was even stronger when comparing the obese group to the athletes, where 18% of mitochondrial proteins were changed, but ~80% were changed less than 1.5-fold. Gene set enrichment revealed a highly significant general trend following the difference in insulin sensitivity (measured by the rate of glucose disappearance –Rd- using a stable isotope – [6,6-$^2$H$_2$]glucose – during a hyperinsulinemic-euglycemic clamp; *Perreault et al., 2018*), where TCA cycle and respiratory electron transport and complex I biogenesis were enriched as follows: athletes > lean > obese > T2D (*Source data 10*). Of note, the T2D group had an enrichment of mitochondrial translation compared to the obese group.

To further explore the relationship between mitochondria and insulin sensitivity, the mitochondrial proteome was correlated to the muscle insulin sensitivity measured using $^2$H$_2$ glucose Rd. As a group, all detected proteins within CI of the ETC were highly correlated with muscle insulin sensitivity (p=4e-15) (*Figure 6E*) and to a lesser extent proteins within CIV (p=0.08) and CV (p=0.09; *Figure 5—figure supplement 1E*). The abundance of CII and CIII, together with the small and large mitochondrial ribosome subunits, was not associated with insulin sensitivity across all the samples (*Figure 5—figure supplement 1F*). Next, we determined the association between the mitochondrial proteome and the levels of C18:0 ceramide in the mitochondria/ER fraction. In line with our previous observations, as a group, CI proteins were inversely correlated with mitochondrial ceramides (p=8e-14) and no association was observed between CII and C18:0 ceramides across samples (*Figure 6F*). Furthermore, components of CIV were also negatively correlated with mitochondrial ceramides although to a lesser extent (*Figure 5—figure supplement 1F*, p=0.026) and CV was not associated with mitochondrial ceramides (*Figure 5—figure supplement 1F*, p=0.737). According to these results, ETC subunits

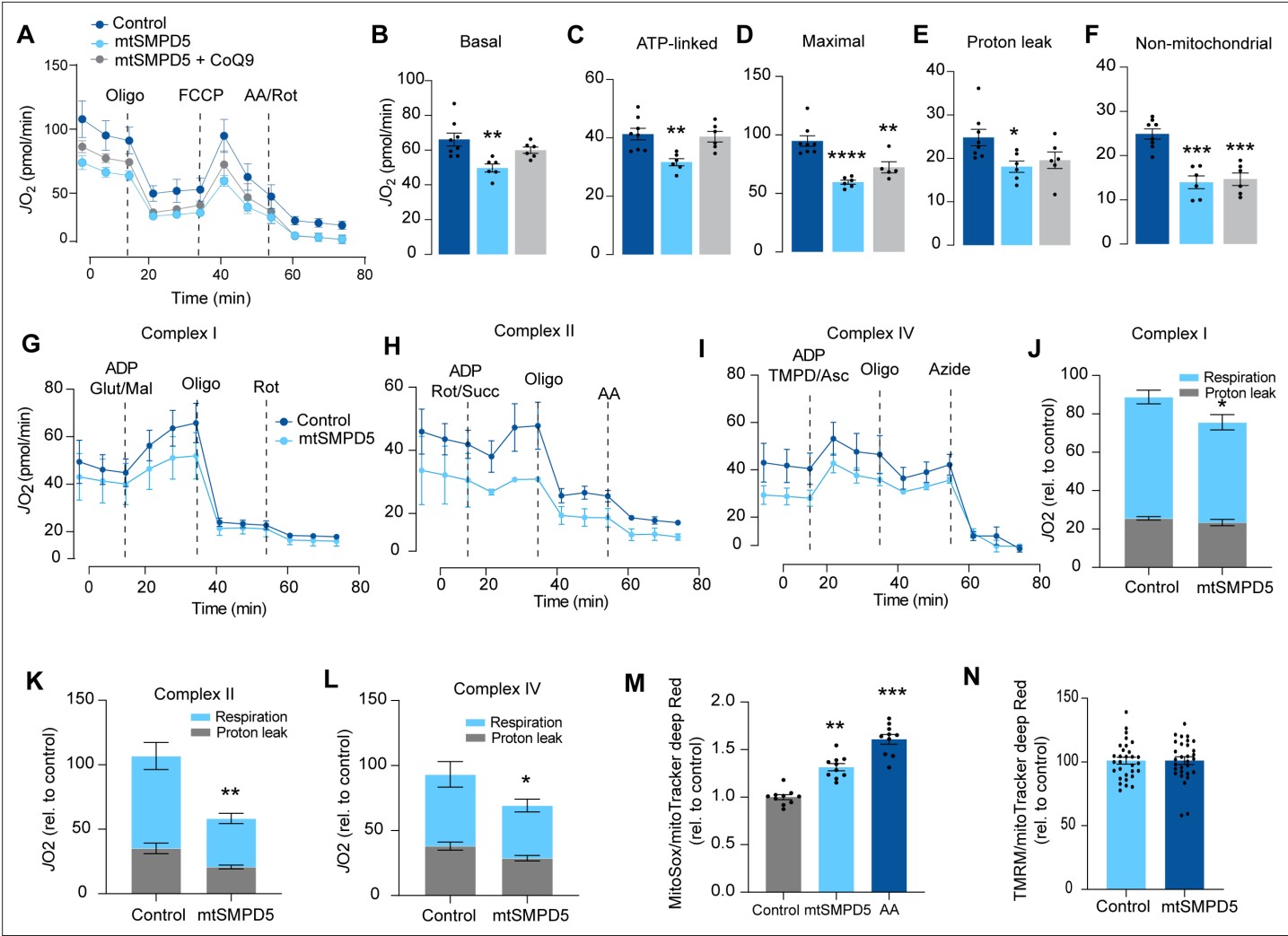

**Figure 5.** Mitochondrial ceramides impair mitochondrial function. (**A**) mtSMPD5 overexpression decreases oxygen consumption rate ($JO_2$) (means ± SEM; n = 6–8 biological replicates) measured by mitochondrial stress test. After 1 hr no $CO_2$ environment, cells were stimulated with oligomycin (Oligo), carbonyl cyanide-p-trifluoromethoxyphenylhydrazon (FCCP), and antimycin A (AA) and Rotenote (Rot) at indicated time points (means ± SEM; n = 6–8 biological replicates). (**B–F**) Quantification of $JO_2$ measured by mitochondrial stress test from **Figure 6A** as described in the 'Materials and methods' section (means ± SEM; n = 6–8 biological replicates). * p<0.05, ** p<0.01, *** p<0.001 vs Control. (**G–L**) mtSMPD5 overexpression diminishes respiratory CI (**G, J**), CII (**H, K**), and CIV (**I, L**). $JO_2$ was performed in permeabilized cells supplemented with adenosine diphosphate (ADP) and CI to IV substrates (means ± SEM; n = 3 biological replicates). Mal, malate; Glut, glutamate; Rot, Rotenone; Succ, succinate; TMPD, tetramethyl-phenylenediamine; Asc, ascorbic acid; Oligo, oligomycin. (**J**), (**K**), and (**L**) are quantifications from graphs (**G**), (**H**), and (**I**), respectively. * p<0.05, ** p<0.01 vs Control. (**M**) mtSMPD5 overexpression increased mitochondrial oxidative stress. Cells were loaded with the redox-sensitive dye MitoSOX and the mitochondrial marker MitoTracker Deep Red for 30 min before imaging in a confocal microscope (see 'Materials and methods') (means ± SEM; n = 10 biological replicates). AA, antimycin A. ** p<0.01, *** p<0.001 vs Control. (**N**) mtSMPD5 overexpression does not alter mitochondrial membrane potential. Cells were loaded with the potentiometric dye tetramethylrhodamine, ethyl ester, perchlorate (TMRM⁺) in non-quenching mode and the mitochondrial marker MitoTracker Deep Red for 30 min before imaging in a confocal microscope (see 'Materials and methods') (means ± SEM; n > 10 biological replicates).

The online version of this article includes the following figure supplement(s) for figure 5:

**Figure supplement 1.** Proteomic analysis of mitochondria isolated from human muscle biopsies.

exhibit differential sensitivity to mitochondrial ceramides, with CI subunits being the most sensitive in human muscle. To uncover structural changes in CI that could correlate with increased ceramide, we mapped those proteins significantly associated with mitochondrial ceramides to the bovine CI structure (**Zhu et al., 2016**). Consistent with L6-mtSMPD5 myotubes, the N and Q modules were the regions with the most negative associated subunits with mitochondria ceramides in human muscle (**Figure 6G**). To determine the conservation in the changes in the mitochondrial proteome induced by

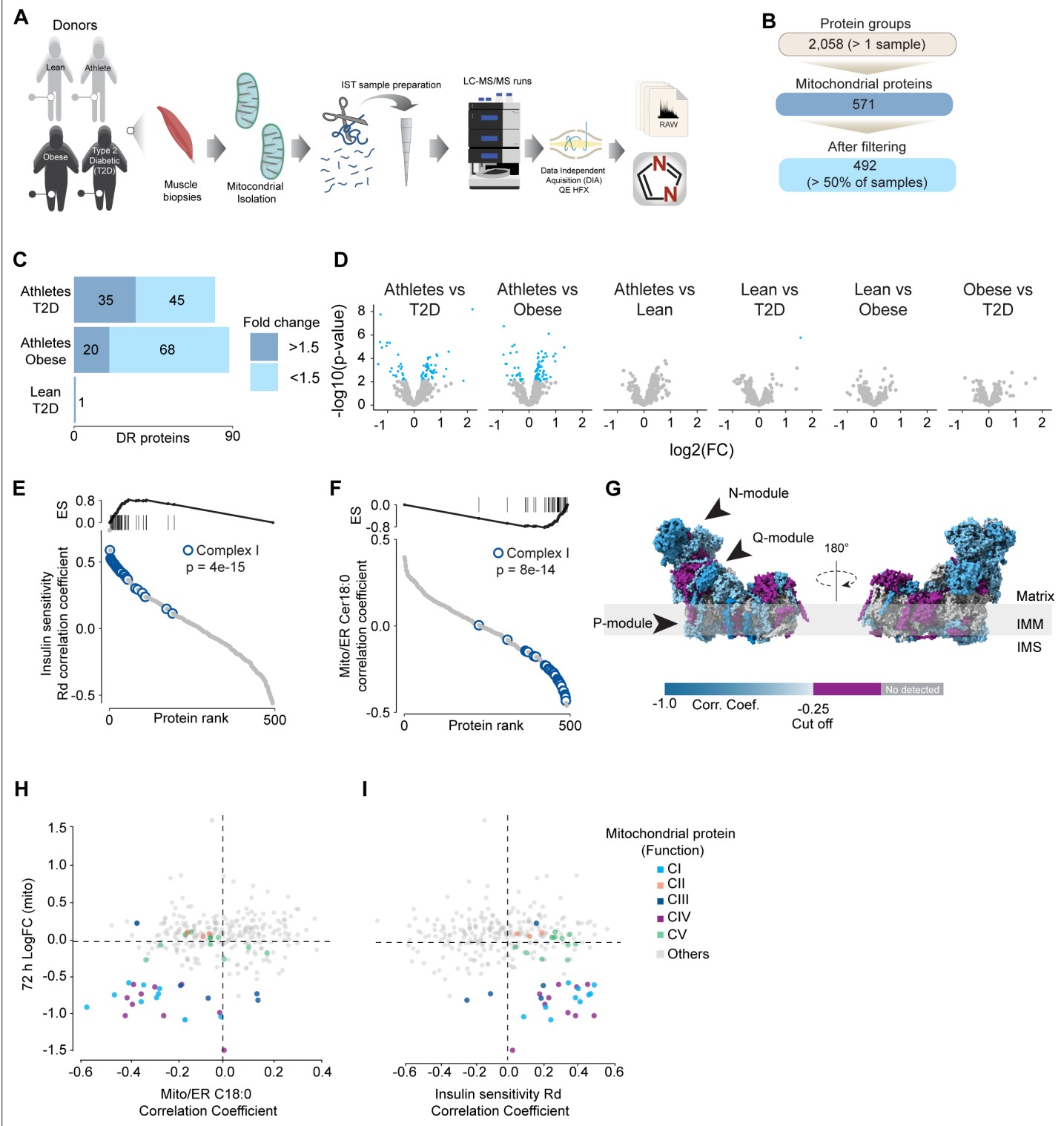

**Figure 6.** Mitochondrial proteome profiling associates complex I with muscle insulin sensitivity. (**A**) Quantification of proteins across samples. (**B**) Number of significant deferential regulated (DR) proteins by pairwise comparison. Groups not shown have no significantly regulated proteins after correcting for multiple testing. (**C**) Gene set enrichment between all comparisons. (**D**) Relative protein abundance in isolated mitochondria from human skeletal muscle cells. Comparisons are shown on the top of each graph. Light blue, significantly regulated proteins (-log10(p-val)=2). (**E**) Proteins rank against rate glucose disappearance during clamp (Rd) correlation. Proteins within complex I of the electron transport chain are highly significant with Rd. (**F**) Proteins rank against mitochondrial ceramide (Cer) 18:0 abundance. Proteins within complex I of the electron transport chain are highly significant with Cer18:0. (**G**) Subunit levels associated with mitochondrial ceramides mapped to the complex I structure (*Zhu et al., 2016*). The colors were

*Figure 6 continued on next page*

*Figure 6 continued*

calculated with an in-house Python script, and the resultant model was rendered using ChimeraX. Gray, no detected; purple, below cutoff (Corr. Coef. = –0.25 to 0.25). (**H**) Quantification of the mitochondrial proteins generated by the summed abundance of all subunits associated with a specific function from mtSMPD5-L6 myotubes after 72 hr vs summed mitochondrial proteins (function) associated with either C18:0 ceramide (left panel) or insulin sensitivity (right panel) from human samples.

increased ceramides, we compared the proteomes of mtSMPD5-L6 myotubes (72 hr after induction) and human muscle biopsies. We observed that across the two datasets CI and CIV subunits were downregulated after mtSMPD5 overexpression and were negatively associated with C18 ceramides in human samples (*Figure 6H*). In turn, CI and CIV were positively associated with muscle insulin sensitivity (*Figure 6I*), suggesting that these ETC subunits exhibited a conserved sensitivity to ceramide accumulation with a potential role in insulin sensitivity.

## Discussion

Insulin resistance is characterized by attenuated insulin-dependent glucose uptake in relevant target tissues, such as muscle and fat, and it plays a central role in cardiometabolic diseases (*James et al., 2021*). In skeletal muscle, mitochondrial ceramides have been linked to insulin resistance (*Perreault et al., 2018*); however, to date, no direct link connecting mitochondrial ceramides with insulin resistance has been established. Furthermore, CoQ depletion and defective mitochondria have also been independently associated with insulin resistance (*Fazakerley et al., 2018a*; *Holland and Summers, 2008*). In the current study, we present evidence suggesting that these factors are mechanistically linked inside mitochondria. Our data demonstrate that increased mitochondrial ceramides are both necessary and sufficient to induce insulin resistance in skeletal muscle. This is likely a function of increased ROS production that results from the specific depletion of the OXPHOS subunits and the concomitant loss of CoQ. Analysis of the human muscle mitochondrial proteome strongly supports mitochondrial ceramide-linked changes in the OXPHOS machinery as major drivers of insulin sensitivity. In this study, we mainly utilized L6 myotubes, which share many important characteristics with primary muscle fibers. Both types of cells exhibit high sensitivity to insulin and respond similarly to maximal doses of insulin, with GLUT4 translocation stimulated between 2 and 4 times over basal levels in response to 100 nM insulin (as shown in *Figures 1–4*, *Sánchez-Aguilera et al., 2018*; *Rosales-Soto et al., 2020*). Additionally, mitochondrial respiration in L6 myotubes has a similar sensitivity to mitochondrial poisons, as observed in primary muscle fibers (as shown in *Figure 5*, *Valladares et al., 2018*). Finally, inhibiting ceramide production increases CoQ levels in both L6 myotubes and adult muscle tissue (as shown in *Figures 2 and 3*). Therefore, L6 myotubes possess the necessary metabolic features to investigate the role of mitochondria in insulin resistance, and this relationship is likely applicable to primary muscle fibers.

Many stressors, including chronic inflammation and anticancer drugs, stimulate endogenous ceramide generation (*Chalfant et al., 2002*) and CoQ depletion in mitochondria (*Tesfay et al., 2019*; *Morad et al., 2012*). Nevertheless, experimental evidence testing the link between these molecules has been lacking. We observed that increased mitochondrial ceramides drive a depletion of mitochondrial CoQ leading to insulin resistance (*Figures 1 and 2*), and that reducing mitochondrial ceramide protects against the loss of CoQ and IR (*Figure 3*). These findings align with our earlier observations demonstrating that mice exposed to HFHSD exhibit mitochondrial CoQ depletion in skeletal muscle (*Fazakerley et al., 2018a*). Given that CoQ supplementation is sufficient to overcome ceramide-induced IR (*Figures 1 and 2*), but a reduction in mitochondrial ceramide does not overcome a loss of CoQ (*Figure 3*), our data support a pathway whereby an increase in mitochondrial ceramides precedes loss of CoQ. Interestingly, inhibition of CoQ synthesis also increased ceramides, suggesting a bidirectionality to the ceramide–CoQ nexus. That said, this effect was modest (*Figure 1*) and we cannot exclude off-target effects of the inhibitor. It is possible that CoQ directly controls ceramide turnover (*Fernández-Ayala et al., 2000*) or alternatively that CoQ inside mitochondria is necessary for fatty acid oxidation (*Frerman, 1988*) and CoQ depletion triggers lipid overload in the cytoplasm promoting ceramide production (*Koves et al., 2008*). Further studies will be needed to determine how CoQ depletion promotes ceramide accumulation.

Our proteomics analysis revealed that the loss of CoQ parallels a loss of mitochondrial ETC complexes CI, CIII, and CIV. These are known to form supercomplexes or respirasomes, where ~25–35% of CoQ are localized in mammals (*Schägger and Pfeiffer, 2000*, *Calvo et al., 2020*). This bulk downregulation of the respirasome induced by ceramides may lead to CoQ depletion. The observation that both palmitate and SMPD5 overexpression trigger CoQ depletion without additive effects supports the notion that ceramides may trigger the depletion of a specific CoQ9 pool localized within the IMM. Despite the significant impact of ceramide on mitochondrial respiration, we did not observe any indications of cell damage in any of the treatments, suggesting that our models are not explained by toxicity and increased cell death (*Figure 1—figure supplement 2H and J*). While the physiological role of respirasomes is still a subject of discussion, it has been suggested that they may enhance energy generation by optimizing electron flow while reducing production of ROS (*Lapuente-Brun et al., 2013*; *Maranzana et al., 2013*), and, therefore, their loss can be predicted to increase ROS generation. The two major mechanisms that might account for the loss of the respirasomes are decreased synthesis or increased degradation. Proteomics data suggests no deficiency in the OXPHOS biosynthetic machinery or assembly proteins and an increase in the machinery for protein import (*Figure 4*). In addition, the absence of mRNA downregulation in mtSMPD5-overexpressing cells strongly suggests that at least a portion of the observed protein depletion within CI is attributed to diminished protein stability. It therefore seems reasonable to speculate that the loss of these mitochondrial complexes is driven by increased degradation. Interestingly, pharmacological CIII inhibition leads to respirasome degradation via oxidative stress produced by reverse electron transfer (RET) (*Guarás et al., 2016*). Since ceramides can directly inhibit CIII (*Gudz et al., 1997*), it is possible that a similar mechanism mediates the effect of ceramides on the respirasome (*Figure 5*). This suggests that defective respirasome activity (e.g., induced by ceramides) triggers ROS, which over time depletes respirasome subunits and a stoichiometric CoQ depletion, leading to further ROS production as a consequence. Another possibility is that, because of its highly hydrophobic nature, ceramides impact membrane fluidity promoting a gel/fluid phase transition (*Pinto et al., 2011*). These alterations in membrane fluidity could decrease respirasome stability. It is likely that bound lipids stabilize the interactions between the complexes in the respirasome and that this is impaired by ceramides. In fact, bound lipid molecules are observed in the structure of the porcine respirasome (*Wu et al., 2016*) and the isolated bovine CI (*Wu et al., 2016*) but none of the lipids identified thus far directly bridge different complexes. In order to understand the role of lipids in stabilizing respirasomes and the role of ceramides in such stabilization, higher-resolution structures will be required (*Mileykovskaya and Dowhan, 2014*).

The current studies pose a number of key unanswered questions. First, how does ceramide accumulate in mitochondria in insulin resistance? This could involve transfer from a different subcellular compartment (*Babiychuk et al., 2011*; *Babiychuk et al., 2008*) or in situ mitochondrial ceramide synthesis. Consistent with the latter, previous studies have suggested that various enzymes involved in ceramide metabolism are specifically found in mitochondria (*Wu et al., 2010*; *Bionda et al., 2004*; *Novgorodov et al., 2011*; *Birbes et al., 2001*; *Yu et al., 2007*; *García-Ruiz et al., 1997*). Notably, CerS1-derived ceramide induces insulin resistance in skeletal muscle (*Turpin-Nolan et al., 2019*). Although this enzyme has not been reported as a mitochondrial protein, it can be transferred from the endoplasmic reticulum surface to the mitochondria under cellular stress in metabolically active tissues such as muscle and brain (*Oleinik et al., 2019*). This provides a potential mechanism where cellular stress, like nutrient overload, may induce transfer of CerS1 to mitochondria, increasing mitochondrial ceramide to trigger insulin resistance.

A further question is how ceramide regulates insulin sensitivity. We present evidence that mitochondrial dysfunction precedes insulin resistance. However, previous studies have failed to observe changes in mitochondrial morphology, respiration, or ETC components during early stages of insulin resistance (*Diaz-Vegas et al., 2020*; *Bonnard et al., 2008*). However, in many cases such studies fail to document changes in insulin-dependent glucose metabolism in the same tissue as was used for assessment of mitochondrial function. This is crucial because we and others do not observe impaired insulin action in all muscles from high-fat fed mice for example (*Fazakerley et al., 2018a*; *Hoehn et al., 2009*). In addition, surrogate measures such as insulin-stimulated Akt phosphorylation may not accurately reflect tissue-specific insulin action as demonstrated in *Figure 1C*. Thus, further work is required to clarify some of these inconsistencies. We observed that mitochondrial ceramides

were associated with the loss of CoQ, increased production of mitochondrial ROS, and impaired mitochondrial respiration (*Fazakerley et al., 2018a*; *Hoehn et al., 2009*; *Fazakerley et al., 2018b*). As discussed above, this is likely a direct result of respirasome depletion. The molecular linkage between ROS production and IR remains unknown. Early studies suggested that ceramides and ROS impaired canonical insulin signaling (*Summers et al., 1998*; *Schubert et al., 2000*; *Powell et al., 2003*); however, our current data do not support this, with the caveat that these were static signaling measures. One possibility is the release of a signaling molecule from the mitochondria that impairs insulin action (*Picard and Shirihai, 2022*). The mitochondrial permeability transition pore (mPTP) is an attractive candidate for this release since its activity is increased by mitochondrial ROS (*Zorov et al., 2014*) and ceramides (*Siskind et al., 2002*). It has been shown that mPTP inhibition protects against insulin resistance in either palmitate- or ceramide-induced L6 myotubes and mice on an HFD (*Taddeo et al., 2014*). Furthermore, mPTP deletion in the liver protects against liver steatosis and insulin resistance in mice (*Cho et al., 2017*). Strikingly, CoQ is an antioxidant and also an inhibitor of mPTP, suggesting that part of the protective mechanism of CoQ may involve the mPTP (*Walter et al., 2000*). Because CoQ can accumulate in various intracellular compartments, it is important to consider that its impact on insulin resistance might be due to its overall antioxidant properties rather than being limited to a mitochondrial effect. Excitingly, mtSMPD5 increased the abundance of mPTP-associated proteins, suggesting a role of this pore in ceramide-induced insulin resistance (*Figure 4—figure supplement 1I*). It is also possible that ceramides generated within mitochondria in SMPD5 cells leak out from the mitochondria into other membranes (e.g., PM and GLUT4 vesicles) affecting other aspects of GLUT4 trafficking and insulin action. However, the observation that ASAH1 overexpression reversed IR without affecting whole-cell ceramides argues against this possibility.

Ultimately, the significant challenge for the field is the discovery of the unknown factor(s) released from mitochondria that cause insulin resistance, their molecular target(s), and the transduction mechanism(s).

The observations described above led us to speculate on whether there is a teleological reason for why these mitochondrial perturbations occur and why they drive insulin resistance. Under conditions of stress, nutrient incorporation into the cell needs to be adjusted to keep the balance between energy supply and utilization. In situations where the mitochondrial respirasome is depleted, the mitochondria's ability to oxidize nutrients can be easily overwhelmed without a corresponding reduction in nutrient uptake. In this scenario, insulin resistance may be a protective mechanism to prevent mitochondrial nutrient oversupply (*Hoehn et al., 2009*). Beyond nutrient uptake, the respirasome depletion could also affect the ability of the mitochondria to switch between different energy substrates depending on fuel availability, named 'metabolic inflexibility' (*Galgani and Fernández-Verdejo, 2021*); this mechanism may potentially play a role in the ectopic lipid accumulation seen in individuals with obesity, a condition linked with cardiometabolic disease.

In summary, our results provide evidence for the existence of a mechanism inside mitochondria connecting ceramides, mitochondrial respiratory complexes, CoQ, and mitochondrial dysfunction as part of a core pathway leading to insulin resistance. We identified that CoQ depletion links ceramides with insulin resistance and define the respirasome as a critical connection between ceramides and mitochondrial dysfunction. While many pieces of the puzzle remain to be solved, identifying the temporal link between ceramide, mitochondrial dysfunction, and CoQ in mitochondria is an important step forward in understanding insulin resistance and other human diseases affecting mitochondrial function.

## Lead contact

Further information and requests for resources and reagents should be directed to and will be fulfilled by the lead contacts, David James (David.james@sydney.edu.au) or James Burchfield (James.burchfield@sydney.edu.au).

## Materials availability

This study generated two new molecular tools to overexpress mitochondria-targeted SMPD5 and ASAH1. The plasmids are available upon request.

## Materials and methods

### Administration of P053 to mice

Male mice of the C57BL/6J strain were obtained from the Animal Resources Centre of Perth (WA, Australia). Mice were housed in a controlled 12:12 hr light–dark cycle, and they had ad libitum access to water and food. The oral gavage administration of P053 (5 mg/kg) was performed daily, while the control animals received vehicle (2% DMSO). The experiments were approved by the UNSW animal care and ethics committee (ACEC 15/48B) and followed guidelines issued by the National Health and Medical Research Council of Australia.

### Cell lines

Mycoplasma-free L6 myotubes overexpressing HA-GLUT4 and HeLa cell lines were used for all in vitro experiments (detailed below each legend). HA-GLUT4 overexpression is essential for studying insulin sensitivity in vitro as we described previously (*Hoehn et al., 2009*). L6-myoblast and HeLa cells were cultured in Dulbecco's Modified Eagle Medium (DMEM) (Gibco by Life Technologies) supplemented with 10% fetal bovine serum (FBS) (v/v) (Gibco by Life Technologies) and 2 mM GlutaMAX (Gibco by Life Technologies) at 37°C and 10% $CO_2$. L6 myoblasts were differentiated in DMEM/GlutaMAX/2% horse serum as described previously (*Hoehn et al., 2009*). The media were replaced every 48 hr for 6 d. For induction of SMPD5 or ASAH1, L6 myotubes were incubated with doxycycline from day 3 until day 6 after the initiation of differentiation. L6 myotubes were used on day 7 after the initiation of differentiation. At least 90% of the cells were differentiated prior to experiments.

### Method details

#### Lentiviral transduction

Lentivirus was made by transfecting LentiX-293T (Takara Bio) cells with Lenti-X Packaging Single Shot (Takara Bio) with one of the following plasmids pLVX-Tet3G, pLVX-TRE3G-SMPD5-Myc-DDK, or pLVX-TRE3G-ASAH-Myc-DDK according to the manufacturer's specifications. Virus-containing media were collected from the LentiX-293T cells and concentrated using Lenti-X Concentrator (Takara Bio). pLVX-Tet3G virus and polybrene were added to L6-myoblast cells and cells were positively selected using neomycin to create Tet3G-expressing cells. The Tet3G-expressing L6 cells were then subsequently infected with polybrene and either the pLVX-TRE3G-SMPD5-Myc-DDK or pLVX-TRE3G-ASAH-Myc-DDK virus. Cells were selected using puromycin to create a Tet-inducible SMPD5-Myc-Flag-DDK or ASAH-Myc-Flag-DDK L6 cell line.

#### Lipid extraction

Two-phase extraction of lipids from frozen tissue samples (20 mg) or cells was carried out using the methyl-tert-butyl ether (MTBE)/methanol/water (10:3:2.5, v/v/v) method (*Matyash et al., 2008*). Frozen tissue samples (~20 mg) were homogenized in 0.2 mL methanol (0.01% butylated hydroxytoluene [BHT]) using a Precellys 24 homogenizer and Cryolys cooling unit (Betin Technologies) with CK14 (1.4 mm ceramide) beads. HeLa cells and L6 myotubes were washed with PBS and scraped into 0.6 mL of ice-cold methanol (*Turner et al., 2018*). Mitochondrial pellets were washed twice to remove BSA from the fraction (see below), and 30 ug of mitochondrial protein was used for extraction. The homogenates were spiked with an internal standard mixture (2 nmole of 18:1/15:0 d5-diacylglycerol and 18:1/17:0 SM, 4 nmole 14:0/14:0/14:0/14:0 cardiolipin, 5 nmole d7-cholesterol, 500 pmole 18:1/17:0 ceramide, and 200 pmole d17:1 sphingosine and d17:1 S1P), then transferred to 10 mL screw cap glass tubes. MTBE (1.7 mL) was added and the samples were sonicated for 30 min in an ice-cold sonicating water bath (Thermoline Scientific, Australia). Phase separation was induced by adding 417 µL of mass spectrometry-grade water with vortexing (max speed for 30 s), then centrifugation (1000 × *g* for 10 min). The upper organic phase was transferred into 5 mL glass tubes. The aqueous phase was re-extracted three times (MTBE/methanol/water 10:3:2.5), combining the organic phase in the 5 mL glass tube. The organic phase was dried under vacuum in a Savant SC210 SpeedVac (Thermo Scientific). Dried lipids were resuspended in 500 µL of 80% MeOH/0.2% formic acid/2 mM ammonium formate and stored at –20°C until analysis.

## Lipid quantification

Lipids were quantified by selected reaction monitoring on a TSQ Altis triple quadrupole mass spectrometer coupled to a Vanquish HPLC system (Thermo Fisher Scientific). Lipids were separated on a 2.1 100 mm Waters Acquity UPLC C18 column (1.7 µM pore size) using a flow rate of 0.28 mL/min. Mobile phase A was 0.1% formic acid and 10 mM ammonium formate in 60% acetonitrile/40% water. Mobile phase B was 0.1% formic acid and 10 mM ammonium formate in 90% isopropanol/10% acetonitrile. Total run time was 25 min, starting at 20% B and holding for 3 min, increasing to 100% B from 3 to 14 min, holding at 100% from 14 to 20 min, returning to 20% B at 20.5 min, and holding at 20% B for a further 4.5 min. Ceramides, sphingomyelin, sphingosine, and sphingosine 1-phosphate were identified as the [M+H]+precursor ion, with m/z 262.3 (sphinganine), 264.3 (sphingosine), or 266.3 (sphinganine) product ion, and m/z 184.1 product ion in the case of sphingomyelin. Diacylglycerols (DAGs) were identified as the [M + NH4]+ precursor ion and product ion corresponding to neutral loss of a fatty acid + $NH_3$. Cardiolipins were identified as the [M + H]+ precursor ion and product ion corresponding to neutral loss of a DAG. Cholesterol was detected using precursor m/z 369.4 and product m/z 161.1. TraceFinder software (Thermo Fisher) was used for peak alignment and integration. The amount of each lipid was determined relative to its class-specific internal standard. Lipidomic profiling of skeletal muscle tissue was performed exactly as described (*Turner et al., 2018*).

## Mass spectrometry sample preparation

Isolated mitochondria were defrosted and centrifuged at 4°C at 4000 × *g* for 15 min, and supernatant was removed. The mitochondrial pellet was resuspended in 100 uL 2% SDC in Tris-HCl buffer (100 mM; pH 8.0) and the protein concentration determined by BCA assay. 10 ug of each sample was aliquoted and volume adjusted to 50 uL with milli-Q water, and samples were reduced and alkylated by addition of TCEP and CAA (10 and 40 mM, respectively) at 60°C for 20 min. Once cooled to room temperature, 0.4 ug MS-grade trypsin and Lys-C were added to each sample, and proteins were digested overnight (16 hr) at 37°C. Peptides were prepared for MS analysis by SDB-RPS stage tips. Two layers of SDB-RPS material were packed into 200 µL tips and washed by centrifugation of StageTips at 1000 × *g* for 2 min in a 96-well adaptor with 50 µL acetonitrile followed by 0.2% TFA in 30% methanol and then 0.2% TFA in water. 50 µL of samples were loaded to StageTips by centrifugation at 1000 × *g* for 3 min. Stage tips were washed with subsequent spins at 1000 × *g* for 3 min with 50 uL 1% TFA in ethyl acetate, then 1% TFA in isopropanol, and 0.2% TFA in 5% ACN. Samples were eluted by addition of 60 µL 60% ACN with 5% $NH_4OH_4$. Samples were dried by vacuum centrifugation and reconstituted in 30 µL 0.1% TFA in 2% ACN.

## Mass spectrometry acquisition and analysis

Samples were analyzed using a Dionex UltiMate 3000 RSLCnano LC coupled to a Exploris Orbitrap mass spectrometer. 3 µL of peptide sample was injected onto an in-house packed 75 µm × 40 cm column (1.9 µm particle size, ReproSil Pur C18-AQ) and separated using a gradient elution, with buffer A consisting of 0.1% formic acid in water and buffer B consisting of 0.1% formic acid in 80% ACN. Samples were loaded to the column at a flow rate 0.5 µL/min at 3% B for 14 min, before dropping to 0.3 µL/min over 1 min and subsequent ramping to 30% B over 110 min, then to 60% B over 5 min and 98% B over 3 min and held for 6 min, before dropping to 50% and increasing flow rate to 0.5 µL/min over 1 min. Eluting peptides were ionized by electrospray with a spray voltage of 2.3 kV and a transfer capillary temperature of 300°C. Mass spectra were collected using a DIA method with varying isolation width windows (widths of m/z 22–589) between 350 and 1650. MS1 spectra were collected between m/z 350–1650 m/z at a resolution of 60,000. Ions were fragmented with an HCD collision energy at 30% and MS2 spectra collected between m/z 300–2000 at resolution of 30,000, with an AGC target of 3e5 and the maximum injection time set to automatic. Raw data files were searched using DIA-NN using a library generated from a 16-fraction high pH reverse phase library (*Yang et al., 2012*). The protease was set to Trypsin/P with one missed cleavage, N-term M excision, carbamidomethylation, and M oxidation options on. Peptide length was set to 7–30, precursor range 350–1650, and fragment range 300–2000, and FDR set to 1%.

## Statistical analysis of L6 mitochondrial proteome

Mouse MitoCarta (REF) was mapped to *Rattus norvegicus* proteins using OrthoDB identifiers downloaded from UniProt. The Rat MitoCarta was used to annotate the L6 proteome. Manual scanning of the annotation revealed a number of known mitochondrial proteins not captured using this approach. Proteins were therefore classified as mitochondria if they were annotated by our mouse:rat MitoCarta (380 proteins), contained 'mitochondrial' in the protein name (78 additional proteins; 231 overlap with MitoCarta), or if the first entry under UniProt 'Subcellular location' was mitochondria (97 additional proteins; 374 overlap with MitoCarta or protein name). LFQ intensities were log 2 transformed and normalized to the median of the mitochondrially annotated proteins. Identification of differentially regulated proteins was performed using moderated *t*-tests (*Smyth, 2004*). Functional enrichment was performed using the STRING web-based platform (*Szklarczyk et al., 2019*).

## Statistical analysis of human proteome and mito-ER lipidome

Analysis of the human proteome and mito-ER lipidome was performed with R (version 4.2.1). Identification of differentially regulated proteins between each group was performed using the R package limma (*Ritchie et al., 2015*), and p-values were corrected with p.adjust (method = "fdr") within each comparison. Correlations were calculated with biweight midcorrelations from the R package WGCNA (*Langfelder and Horvath, 2008*). Gene set enrichment was performed with the R package clusterProfiler (*Wu et al., 2021*) utilizing pathways from Reactome for differentially regulated proteins (*Fabregat et al., 2018*). Custom mitochondrial genes were constructed from HGNC Database (*Seal et al., 2023*), and enrichment and enrichment figures were done with the R package fgsa (https://www.biorxiv.org/content/10.1101/060012v3).

## High pH reverse-phase fractionation and library generation

A pooled sample was made by combining 1 uL of each sample and fractionated by high pH reverse-phase liquid chromatography. 50 uL of pooled sample was injected onto a Waters XBridge Peptide BEH C18 column (4.6 × 250 mm, 130 Å, 3.5 um) using a Thermo Scientific UltiMate 3000 BioRS System and peptides separated using gradient elution at 1 mL/min, with the column oven set to 30°C. Buffer A consisted of 10 mM ammonium formate, and buffer B consisted of 10 mM ammonium formate in 80% acetonitrile, which both adjusted to pH 9.0 with ammonium hydroxide. Initially buffer B was set to 10% and ramped up to 40% over 11 min, before ramping up to 100% B over 1 min and held for 5 min before returning to 10% for re-equilibration. Peptides were separated into 64 fractions collected between 3.45 min and 14.5 min, and samples were concatenated into 16 final fractions. Fractions were dried using a GeneVac 2.0 vacuum centrifuge using the HPLC program, with a maximum temperature of 60°C. Fractions were resuspended in 10 uL 0.1% TFA in 2% ACN and 2 uL was injected and separated as described for the DIA samples above; however, MS was acquired in a DDA manner. An MS1 was collected between m/z 350–1650 with a resolution of 60,000. The top 15 most intense precursors were selected from fragmentation with an isolation window of 1.4 m/z, resolution of 15,000, HCD collision energy of 30%, with an exclusion window of 30 s. Raw files were searched with MaxQuant against a FASTA file containing the reviewed UniProt human proteome (downloaded May 2020).

## Matrigel-coated plates

Matrigel diluted 1:100 v/v in ice-cold PBS was dispensed into 96-well plates (Eppendorf Cell Imaging plate, UNSPSC 41122107; and Perkin Elmer Cell Carrier Ultra, Cat# 6055300) and incubated for 2 hr at room temperature. Before use, plates were washed twice in PBS at room temperature.

## HA-GLUT4 assay

HA-GLUT4 levels on the PM were determined as described previously (*Hoehn et al., 2009*; *Govers et al., 2004*). L6 myotubes stably overexpressing HA-Glut4 were washed twice with warm PBS and serum-starved for 2 hr (in DMEM/0.2% BSA/GlutaMAX/with 220 mM bicarbonate [pH 7.4] at 37°C, 10% $CO_2$). Cells were then stimulated with insulin for 20 min, after which the cells were placed on ice and washed three times with ice-cold PBS. Cells were blocked with ice-cold 10% horse serum in PBS for 20 min, fixed with 4% paraformaldehyde (PFA) for 5 min on ice and 20 min at room temperature. PFA was quenched with 50 mM glycine in PBS for 5 min at room temperature. We measured the

accessibility of the HA epitope to an anti-HA antibody (Covance, 16B12) for 1 hr at room temperature. Cells were then incubated with 20 mg/mL goat anti-mouse Alexa-488-conjugated secondary antibody (Thermo Fisher Scientific) for 45 min at room temperature. The determination of total HA-GLUT4 was performed in a separate set of cells following permeabilization with 0.01% saponin (w/v) and anti-HA staining (as above). Each experimental treatment group had its own total HA-GLUT4. A FLuostar Galaxy microplate reader (BMG LABTECH) was used to measure fluorescence (excitation 485 nm/ emission 520 nm). Surface HA-GLUT4 was expressed as a fold over control insulin condition.

## Induction of insulin resistance

To promote insulin resistance, cells were stimulated for 16 hr with 150 μM palmitate-BSA or EtOH-BSA as control. The palmitate was complexed with BSA as described previously (*Hoehn et al., 2009*). Briefly, fatty acid was dissolved in 50% ethanol and then diluted 25 times in 10.5% fatty acid-free BSA solution. These stock solutions were further diluted in culture media to reach a final concentration of 150 μM (final lipid:BSA ratio 4:1).

## Glycogen synthesis assay

L6 myotubes overexpressing ASAH were grown and differentiated in 12-well plates, as described in the 'Cell lines' section, and stimulated for 16 hr with palmitate-BSA or EtOH-BSA, as detailed in the 'Induction of insulin resistance' section.

On day 7 of differentiation, myotubes were serum-starved in plain DMEM for three and a half hours. After incubation for 1 hr at 37°C with 2 μCi/mL D-[U-14C]-glucose in the presence or absence of 100 nM insulin, glycogen synthesis assay was performed, as described previously (*Zarini et al., 2022*).

## Coenzyme Q determination

CoQ9 and CoQ10 content in cell lysates and mitochondrial fractions were determined as described previously (*Burger et al., 2020*). Aliquots of 15 μg mitochondrial protein as prepared below were adjusted to a volume of 100 μL with water and subsequently mixed with 250 μL ice-cold methanol containing 0.1% HCl, 20 μL internal standard (CoQ8, 200 pmol in hexane, Avanti Polar Lipids), and 300 μL of hexane. The mixture was vortexed for 30 s, centrifuged ($9000 \times g \times 5$ min), and the supernatant was transferred into deepwell plate 96/1000 uL (Cat# 951032905). Samples were dried using a rotary evaporator (GeneVac, low BP at 45°C for 40 min). The resulting dried lipids were re-dissolved in 100 uL of 100% EtOH (HPLC grade), transferred into HPLC vials, and stored at –20°C until analysis by LC/MS.

LC-MS/MS was performed on a Vanquish LC (Thermo Fisher) coupled to a TSQ Altis triple quadrupole mass spectrometer (Thermo Fisher Scientific). Samples were kept in the autosampler at 4°C and 15 μL was injected on onto column (50 × 2.1 mm, Kinetex 2.6 μm XB-X18 100A) at 45°C, and CoQ8, CoQ9, and CoQ10 were separated by gradient elution using mobile phase A (2.5 mM ammonium formate in 95:5 methanol:isopropanol) and mobile phase B (2.5 mM ammonium formate in 100% isopropanol) at 0.8 mL/min. An initial concentration of 0% B was held for 1 min before increasing to 45% B over 1 min and held for 1 min, before decreasing back to 0% B over 0.5 min and column re-equilibrated over 1.5 min. Under these conditions, CoQ8 eluted at 1.0 min, CoQ9 at 1.6 min, and CoQ10 at 2.0 min. Eluent was then directed into the QqQ with a source voltage of 3.5 kV, sheath gas set to 2, auxiliary gas set to 2, and a transfer capillary temperature of 350°C. Ammonium adducts of each of the analytes were detected by SRM with Q1 and Q3 resolution set to 0.7 FWHM with the following parameters: [CoQ8 + NH4]+, m/z 744.9 197.1 with collision energy 32.76; [CoQ9 + NH4]+, m/z 812.9 197.1 with collision energy 32.76; [CoQ9H2 + NH4]+, m/z 814.9 197.1 with collision energy 36.4; and [CoQ10 + NH4]+, m/z 880.9 197.1 with collision energy 32.76. CoQ9 and CoQ10 areas were normalized to the internal standard CoQ8 levels (20 ng/mL). CoQ9 and CoQ10 were quantified against external standard curves generated from authentic commercial standards obtained from Sigma-Aldrich (USA).

## Mitochondrial isolation

Mitochondrial isolation from cultured L6 myotubes was performed as described elsewhere (*Bui et al., 2010*; *Frezza et al., 2007*). Briefly, cells were homogenized in an ice-cold mitochondrial isolation

buffer (5 mM HEPES, 0.5 mM EGTA, 200 mM mannitol, and 0.1% BSA, pH 7.4 containing protease inhibitors) using a Cell Homogenizer with 18 micron ball. Cells were passed through the Cell Homogenizer 10 times using 1 mL syringe. Cell Homogenizer was equilibrated with 1 mL of ice-cold isolation buffer prior to the experiment. Homogenates were centrifuged at $700 \times g$ for 10 min and the supernatant centrifuged at $10,300 \times g$ for 10 min to generate the crude mitochondrial pellet. The $10,300 \times g$ pellet was resuspended in 1 mL of isolation buffer and transferred into a polycarbonate tube containing 7.9 mL of 18% Percoll in the homogenization buffer and centrifuged at $95,000 \times g$ at 4°C for 30 min. The mitochondrial pellet was collected and diluted in a homogenization buffer (1 mL) and centrifuged at $10,000 \times g$ for 10 min at 4°C. The supernatant was discarded, and the pellet was washed with a homogenization buffer without BSA, followed by protein quantification with BCA protein assay.

Mitochondria from adult skeletal muscle (from mixed hindlimb muscle) were isolated by differential centrifugation as described previously (*Montgomery et al., 2019*). Briefly, muscle was diced in CP-1 medium (100 mM KCl, 50 mM Tris/HCl, pH 7.4, and 2 mM EGTA), digested on ice for 3 min in CP-2 medium (CP-1, to which was added 0.5% [w/v] BSA, 5 mM $MgCl_2$, 1 mM ATP and 2.45 units/mL Protease Type VIII [Sigma P 5380]) and homogenized using an ultra-turrax homogenizer. The homogenate was spun for 5 min at $500 \times g$ and 4°C. The resulting supernatant was subjected to a high-speed spin ($10,600 \times g$, 10 min, 4°C), and the mitochondrial pellet was resuspended in CP-1. The $10,600 \times g$ spin cycle was repeated, the supernatant removed, and the mitochondrial pellet snapped frozen.

## Western blotting

After insulin stimulation or mitochondrial isolation, samples were tip sonicated in 2% SDS-RIPA. Insoluble material was removed by centrifugation at $21,000 \times g \times 10$ min. Protein concentration was determined by bicinchoninic acid method (Thermo Scientific). 10 µg of protein was resolved by SDS-PAGE and transferred to PDVF membranes. Membranes were blocked in Tris-buffered saline (TBS) 4% skim milk for 30 min at room temperature, followed by primary antibody incubation (detailed antibodies are provided in the '*Supplementary file 1*'). Membranes were washed in TBS 0.1% Tween (TBS-T) and incubated with appropriate secondary antibodies (IRDye700- or IRDye800-conjugated) in TBS-T 2% skim milk for 45 min at room temperature. Images were obtained by using 700 or 800 nm channels using Odyssey IR imager. Densitometry analysis of immunoblots was performed using Image Studio Lite (version 5.2). Uncropped western blots are provided in the supplementary material.

## Seahorse extracellular flux analyses

Mitochondrial respiration ($JO_2$) of intact cells were measured using an XF HS mini Analyser Extracellular Flux Analyzer (Seahorse Bioscience, Copenhagen, Denmark). L6 myoblasts were seeded and differentiated in Seahorse XFp culture plates coated with Matrigel and assayed after incubation at 37°C without $CO_2$ for 1 hr. Prior to the assay, cells were washed three times with PBS, once with bicarbonate-free DMEM buffered with 30 mM Na-HEPES, pH 7.4 (DMEM/HEPES), and then incubated in DMEM/HEPES supplemented with 0.2% (w/v) BSA, 25 mM glucose, 1 mM GlutaMAX, and 1 mM glutamine (Media B), for 1.5 hr in a non-$CO_2$ incubator at 37°C. During the assay, respiration was assayed with mix/wait/read cycles of 2/0/2 min for L6 myotubes. Following assessment of basal respiration, the following compounds (final concentrations in parentheses) were injected sequentially: oligoymcin (10 µg/mL), BAM15 (10 mM), and rotenone/antimycin A (5 µM/10 µM). All of these reagents were obtained from Sigma-Aldrich. Basal (baseline – Ant./Rot), ATP-linked respiration (determined by basal – oligomycin), maximal respiration (calculated by FCCP – AntA/Rot), and non-mitochondrial respiration (equal to AntA/Rot) were determined as described previously (*Díaz-Vegas et al., 2018*). Protein concentration was determined immediately after the assay and data are presented as $O_2$/min. Complex specific activity in permeabilized cells was performed according to *Kory et al., 2020*. The cells were seeded and the media changed to a buffer consisting of 70 mM sucrose, 220 mM mannitol, 10 mM $KH_2PO_4$, 5 mM $MgCl_2$, 2 mM HEPES (pH 7.2), 1 mM EGTA, and 0.4% BSA. Then, flux measurements began after taking three baseline measurements. The cells were permeabilized by adding digitonin (1 nM) and 1 mM ADP, followed by injecting respiratory complex substrates or ADP only (complex I, glutamate/malate [5 mM/2.5 mM]; complex II, succinate/rotenone [10 mM/1 µM]; complex III, and complex IV, N,N,N,N-tetramethyl-*p*-phenylenediamine/ascorbate [0.5 mM/2 mM]). Subsequently, oligomycin (1 µg/mL) and respective complex inhibitors were added (complex I, 1 µM

rotenone; complexes II and III, 20 µM antimycin A; complex IV, 20 mM sodium azide). Wells where cells detached from the plate during the assay were excluded from the analysis.

## Mitochondrial membrane potential

Mitochondrial membrane potential was measured by loading cells with 20 nM tetramethylrhodamine, ethyl ester (TMRM+, Life Technologies) for 30 min at 37°C plus MitoTracker Deep Red (MTDR). MTDR was used to normalize the fluorescence among the different mitochondrial populations as reported previously (*Díaz-Vegas et al., 2018*). TMRM+ fluorescence was detected using the excitation-emission $\lambda$ 545–580/590 nm, and MTDR was detected using an ex/em ~644/665 nm using confocal microscopy. The mitochondrial membrane potential was evaluated as raw fluorescence intensity of background-corrected images.

## Mitochondrial oxidative stress

MitoSOX Red was administered as described by the manufacturer (Molecular Probes); at the end of the induction period, cells were washed twice with PBS and incubated with 0.5 µM MitoSOX Red for 30 min plus MTDR. MTDR was used to normalize the fluorescence among the different mitochondrial populations as reported previously reported. Cells were cultured in low-absorbance, black-walled 96-well plates. After MitoSOX treatment, cells were quickly washed with PBS and fluorescence was detected on a confocal microscope. MitoSOX fluorescence was detected using the excitation–emission $\lambda$ 396/610 nm, and MTDR was detected using an ex/em ~644/665 nm using confocal microscopy.

## RT-PCR

RNA was extracted by addition of TRIzol (Thermo Fisher) followed by addition of 0.1× volume of 1-bromo-3-chloropropane. Samples were centrifuged at 13,000 × *g* for 15 min for phase separation. The clear phase was transferred to a fresh tube and an equal volume of isopropanol was added. Samples were centrifuged at 13,000 × *g* for 10 min to precipitate RNA. RNA was washed three times in 70% ethanol with centrifugation. RNA was resuspended in DEPC-treated water and quantified on a NanoDrop 2000 (Thermo Scientific). RNA was reverse-transcribed to cDNA using PrimeScript Reverse Transcriptase (Takara) as per the manufacturer's instructions. qPCR was performed using FastStart SYBR Green MasterMix (2x) (Bio-Rad) as per the manufacturer's instructions. All primer sequences can be found in *Supplementary file 2*.

## Statistical analysis

Data are presented as mean ± SEM. Statistical tests were performed using GraphPad Prism version 9. We employed Student's *t*-test and one-way ANOVA followed by Dunnett's post-test for paired data and multiple comparisons, respectively. HA-GLUT4 and western blot assay was analyzed by using Kruskal–Wallis with Dunn's multiple-comparison test. CoQ and ceramide abundance were analyzed with ordinary one-way ANOVA and Dunnett's multiple-comparison test. Finally, for comparison of two groups (CoQ and ceramides levels in mice), we used Student's *t*-test. Significant effects were defined as $p < 0.05$ by these tests as reported in the figures.

## Acknowledgements

This work was supported by the National Health and Medical Research Council (NHMRC) Project Grants: GNT1120201 and GNT1061122 to DEJ; GNT2013621 to JGB, DEJ, and ADV; and GNT112613 to NT, JM, and AD. BCB received a research grant from Eli Lilly and from NIH R01DK111559. DEJ is an Australian Research Council (ARC) Laureate Fellow. JGB and ADV were supported by the Diabetes Australia Research Program (Y22G-DIAA) and the Mitochondrial Foundation (Mitofoundation, G057). The content is solely the responsibility of the authors and does not necessarily represent the official views of the NHMRC or ARC. The authors also acknowledge the facilities, and the scientific and technical assistance of Sydney Cytometry and the Sydney Mass Spectrometry Facility, at the Charles Perkins Centre, University of Sydney. The authors are extremely grateful to Scott A Summers, William L Holland, and Navdeep S Chandel for their thoughtful discussion of the project.

# Additional information

## Competing interests

Andrew P Ryan, Joseph T Brozinick: is an employee of Eli Lilly. The author has no other competing interests to declare. David E James: Senior editor, eLife. The other authors declare that no competing interests exist.

## Funding

| Funder | Grant reference number | Author |
|---|---|---|
| National Health and Medical Research Council | 2013621 | Alexis Diaz-Vegas<br>James G Burchfield<br>David E James |
| Australian Research Council | DP210102099 | Alexis Diaz-Vegas |
| Australian Research Council | FL200100096 | David E James |
| National Health and Medical Research Council | GNT1120201 | David E James |
| National Health and Medical Research Council | GNT1061122 | David E James |
| National Health and Medical Research Council | GNT112613 | Nigel Turner<br>Jonathan C Morris<br>Anthony S Don |
| Diabetes Australia | Y22G-DIAA | James G Burchfield<br>Alexis Diaz-Vegas |
| Mitochondrial Foundation | Mitofoundation, G057 | James G Burchfield<br>Alexis Diaz-Vegas |

The funders had no role in study design, data collection and interpretation, or the decision to submit the work for publication.

## Author contributions

Alexis Diaz-Vegas, Conceptualization, Data curation, Formal analysis, Funding acquisition, Investigation, Visualization, Methodology, Writing – original draft, Project administration, Writing – review and editing; Søren Madsen, Data curation, Formal analysis, Writing – original draft; Kristen C Cooke, Data curation, Formal analysis, Investigation, Methodology; Luke Carroll, Jasmine XY Khor, Nigel Turner, Xin Y Lim, Miro A Astore, Jonathan C Morris, Anthony S Don, Karin A Zemski Berry, Andrew P Ryan, Bryan C Bergman, Joseph T Brozinick, Investigation, Methodology; Amanda Garfield, Investigation; Simona Zarini, Conceptualization, Investigation, Methodology; David E James, Conceptualization, Supervision, Funding acquisition, Project administration, Writing – review and editing; James G Burchfield, Conceptualization, Resources, Formal analysis, Supervision, Funding acquisition, Investigation, Visualization, Methodology, Writing – original draft, Project administration, Writing – review and editing

## Author ORCIDs

Alexis Diaz-Vegas http://orcid.org/0000-0001-5227-4482
Luke Carroll http://orcid.org/0000-0002-8600-4023
Jonathan C Morris http://orcid.org/0000-0002-5109-9069
Amanda Garfield http://orcid.org/0009-0006-1449-4939
Karin A Zemski Berry https://orcid.org/0000-0002-7089-691X
Andrew P Ryan https://orcid.org/0000-0001-8342-3743
Joseph T Brozinick https://orcid.org/0000-0002-7946-3432
David E James https://orcid.org/0000-0001-5946-5257
James G Burchfield https://orcid.org/0000-0002-6609-6151

## Ethics

This study involving human participants received ethical clearance and authorization from the Colorado Multiple Institution Review Board at the University of Colorado Anschutz Medical Campus. Prior

to their involvement in the study, all participants submitted written informed consent. All information regarding this ethic approval can be found in JCI.

The experiments obtained approval from the UNSW animal care and ethics committee (ACEC 15/48B) and adhered to the guidelines set forth by the National Health and Medical Research Council of Australia. All information regarding this ethic approval can be found in Nat Commun. 2018; 9: 3165.

Reviewer #1 (Public Review): https://doi.org/10.7554/eLife.87340.3.sa1
Reviewer #2 (Public Review): https://doi.org/10.7554/eLife.87340.3.sa2
Author Response https://doi.org/10.7554/eLife.87340.3.sa3

## Additional files

### Supplementary files
- MDAR checklist
- Supplementary file 1. Reagents.
- Supplementary file 2. Sequences.
- Source data 1. Uncropped gels *Figures 2 and 3F*.
- Source data 2. Uncropped gells *Figures 2F and 3*.
- Source data 3. Lipidomic analysis L6.
- Source data 4. Lipidomic analysis Hela.
- Source data 5. Lipidomic analysis L6s with mtSMPD5 overexpression.
- Source data 6. Lipidomic analysis of mice exposed to chow or HFD with or without P051.
- Source data 7. Correlation analysis of lipidomics and proteomics in human samples.
- Source data 8. Correlation analysis between proteomics and human traits.
- Source data 9. Correlation analysis between C18 ceramides and human muscle proteomics.
- Source data 10. Human proteomics PairWiseComp.
- Source data 11. Human proteomics analysis median normalised.
- Source data 12. Human proteomics analysis Rd_clamp_correlation_all_protein.

### Data availability
All Lipidomic and proteomic analyses are available as supporting files. Uncropped western blotting are found in source data files.

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
