## [Editor Report · eLife assessment]

This **important** study highlights a potential connection between fatty acid intrusion into myocytes and increases in mitochondrial ceramide that cause deficits in coenzyme Q and consequent insulin resistance. The authors primarily use the L6 myocyte model, which may not fully recapitulate in vivo conditions; however, the article shows **compelling** data in mice that substantially supports the L6 cell results. Overall, this study provides a strong framework for a **compelling** pathway of myocyte dysfunction and for continued efforts to test the **important** hypotheses that are presented.

---

## [Referee Report · Reviewer #1 (Public Review)]

Previous reports suggested an association between ceramide accumulation in skeletal muscle and disruption of insulin signaling and metabolic dysregulation. Mechanistically, however, how intracellular ceramide attenuates insulin action and reduces metabolism is not fully understood. It was suggested that insulin receptor (IR) signaling to PI3-K/AKT is inhibited by elevated intracellular ceramide. However, other studies failed to demonstrate an inhibitory effect of ceramide on PI3K/AKT. More recently, a study was published describing that intracellular localization of diacylglycerols and sphingolipids influences insulin sensitivity and mitochondrial function in human skeletal muscle (PMID: 29415895). In the present study, Diaz-Vegas and colleagues used an in vitro system to investigate this topic further and better understand how intracellular ceramide accumulation causes cellular insulin resistance and metabolic dysregulations in cultured myocytes.

The authors applied multiple methods to achieve this goal. Among these procedures are:

1. The overexpression of enzymes involved in mitochondrial ceramide synthesis and degradation;

2. Treatments of myocytes with different pharmacological tools to validate their findings;

3. Mitochondrial proteomics and lipidomics analyses.

The effects of these experimental conditions and treatment on intracellular lipids contents, mitochondrial functions, and insulin signaling in myocytes were then evaluated.

Findings:

The author's findings indicate that incubation of myocytes with palmitate increases mitochondrial ceramide and reduces the insulin-stimulated GLUT4-HA translocation to the myocyte surface without affecting AKT activation. The elevation in mitochondrial ceramide lowers the coenzyme Q levels e depletes the electron transport chain (ETC) components, impairing mitochondrial respiration. Such mitochondrial dysfunction appears to attenuate the translocation of GLUT4-HA to the plasma membrane of the L6-myotubule. Also, mitochondrial proteomic analysis revealed an association of insulin sensitivity with mitochondrial ceramide and ETC expression levels in human muscle.

Based on these findings, the authors propose a mechanism whereby the building up of ceramide inside mitochondria depletes CoQ and compromises mitochondrial respiratory complexes, raising ROS. The resulting mitochondrial dysfunction causes insulin resistance in cultured myocytes. They postulate that CoQ depletion links ceramides with insulin resistance and define the respirasome as a critical connection between ceramides and mitochondrial dysfunction.

Relevance and critiques:

This original study provides direct evidence that mitochondrial ceramide accumulation depletes CoQ and downregulates multiple ETC components in myocytes. Consequently, elevation in the levels of reactive oxygen species (ROS) and mitochondrial dysfunctions occur. The authors proposed that such mitochondrial dysregulation attenuates insulin-stimulated GLUT4 translocation to the plasma membrane of L6-myotubules. Moreover, mitochondrial ceramide accumulation does not affect insulin action on AKT activation.

Overall, this is a well-done study, showing that in obesity, elevated mitochondrial ceramide suppresses mitochondrial function and attenuates insulin action on glucose transporter GLUT4 translocation into the myocyte surface. The main conclusion is supported by the results presented. The study also applied multiple methods and described several experiments designed to test the author's central hypothesis.

Importantly, these new findings shed light on possible cellular mechanisms whereby ectopic fat deposition in skeletal muscle drives insulin resistance and metabolism dysregulation. The results demonstrating that alterations in mitochondrial ceramide are sufficient to attenuate insulin-stimulated GLUT4 trafficking in cultured myocytes are very interesting. Well-done.

Comments for further discussion and suggestions:

Although the author's results suggest that higher mitochondrial ceramide levels suppress cellular insulin sensitivity, they rely solely on a partial inhibition (i.e., 30%) of insulin-stimulated GLUT4-HA translocation in L6 myocytes. It would be critical to examine how much the increased mitochondrial ceramide would inhibit insulin-induced glucose uptake in myocytes using radiolabel deoxy-glucose.

Another important question to be addressed is whether glycogen synthesis is affected in myocytes under these experimental conditions. Results demonstrating reductions in insulin-stimulated glucose transport and glycogen synthesis in myocytes with dysfunctional mitochondria due to ceramide accumulation would further support the author's claim.

In addition, it would be critical to assess whether the increased mitochondrial ceramide and consequent lowering of energy levels affect all exocytic pathways in L6 myoblasts or just the GLUT4 trafficking. Is the secretory pathway also disrupted under these conditions?

Additional suggestions:

• Figure 1: How does increased mitochondrial ceramide affect fatty acid oxidation (FAO) in L6-myocytes? As the accumulation of mitochondrial ceramide inhibits respirasome and mitochondrial activity in vitro, can reduced FAO in vivo, due to high mitochondrial ceramide, accounts for ectopic lipid deposition in skeletal muscle of obese subjects?

• Figure 2: Although the authors show that mtSMPD5 overexpression does not affect ceramide abundance in whole cell lysate, it would be critical to examine the abundance of this lipid in other cellular membranes and organelles, particularly plasma membrane. What is the effect of mtSMPD5 overexpression on plasma membrane lipids composition? Does that affect GLUT4-containing vesicles fusion into the plasma membrane, possibly due to depletion of v-SNARE or tSNARE?

• Figure 4: One critical piece of information missing is the effect (if any) of mitochondrial ceramide accumulation on the mRNAs encoding the ETC components affected by this lipid. Although the ETC protein's lower stability may account for the effect of increased ceramide, transcriptional inhibition can't be ruled out without checking the mRNA expression levels for these ETC components.

In the revised version of their study, the authors nicely addressed all concerns previously raised. The amount of work that went into the revisions is appreciated. All weak points have been properly addressed, and the manuscript has improved substantially.

---

## [Referee Report · Reviewer #2 (Public Review)]

Summary

The findings reported by Diaz-Vegas et al. extend those described in a previous paper from the same group establishing a role for mitochondrial CoQ depletion in the development of insulin resistance in muscle and adipose tissue (Fazakerley, 2018). In this new report, investigators sought to determine whether CoQ depletion contributes to insulin resistance caused by palmitate exposure and/or intracellular ceramide accumulation. To this end, researchers employed a widely used in vitro model of insulin resistance wherein L6 myocytes develop impaired Glut4 translocation upon exposure to palmitate (in this case, 150 uM for 16 hours). This model was combined with a variety of pharmacologic and genetic manipulations aimed at augmenting or inhibiting CoQ biosynthesis and/or ceramide biosynthesis, specifically in mitochondria. This series of experiments produced a valuable and provocative body of evidence positioning CoQ depletion downstream of mitochondrial ceramide accumulation and necessary for both palmitate- and ceramide-induced insulin resistance in L6 myocytes. Investigators concluded that mitochondrial ceramides, CoQ depletion and respiratory dysfunction are part of a core pathway leading to insulin resistance.

Strengths

The study provides exciting, first-time evidence linking palmitate-induced insulin resistance to ceramide accumulation within the mitochondria and subsequent depletion of CoQ. Ceramide accumulation specifically in mitochondria was found to be necessary and sufficient to cause insulin resistance in cultured L6 myocytes.

The in vitro experiments featured a set of mitochondrial-targeted genetic manipulations that permitted up/down-regulation of ceramide levels specifically in the mitochondrial compartment. Genetically induced mitochondrial ceramide accumulation led to CoQ depletion, which was accompanied by increased ROS production and diminution of ETC proteins and OXPHOS capacity and impaired insulin action, thereby establishing cause/effect.

Analysis of mitochondria isolated from human muscle biopsies obtained from individuals with disparate metabolic phenotypes revealed a positive correlation between complex I proteins and insulin sensitivity and a negative correlation with mitochondrial ceramide content. While it is likely that many factors contribute to these correlations, the results support the possibility that the ceramide/CoQ mechanism might be relevant to glucose control in humans.

Investigators were responsive to the reviewers' queries and critiques and performed additional experiments to bolster the interpretations and conclusions put forth in the manuscript. These included experiments to confirm that mito-targeted SMPD5 does not cause toxicity in L6 myocytes, and further studies using targeted metabolomic and lipidomic analyses to investigate the impact of ceramide depletion on CoQ levels in mice fed a high-fat diet and treated with P053 (a selective inhibitor of CerS1). The results were consistent with the in vitro findings.

Overall, these important findings offer valuable new insights into mechanisms that connect ceramides to insulin resistance and mitochondrial dysfunction, and could inform new therapeutic approaches towards improved glucose control.

Weaknesses

The mechanistic aspect of the work and conclusions put forth rely heavily on studies performed in cultured myocytes, which are highly glycolytic and generally viewed as an imperfect model for studying muscle metabolism and insulin action. Nonetheless, results from the cell culture model are generally convincing and align with the descriptive data from studies in animal models. Overall, the findings provide a strong rationale for moving this line of investigation into mouse gain/loss of function models.

One caveat of the approach taken is that exposure of cells to palmitate alone is not reflective of in vivo physiology. It would be interesting to know if similar effects on CoQ are observed when cells are exposed to a more physiological mixture of fatty acids that includes a high ratio of palmitate, but better mimics in vivo nutrition.

---

## [Author Response]

The following is the authors’ response to the original reviews.

Assessment note: “Whereas the results and interpretations are generally solid, the mechanistic aspect of the work and conclusions put forth rely heavily on in vitro studies performed in cultured L6 myocytes, which are highly glycolytic and generally not viewed as a good model for studying muscle metabolism and insulin action.”

While we acknowledge that in vitro models may not fully recapitulate the complexity of in vivo systems, we believe L6 myotubes are appropriate for studying the mechanisms underlying muscle metabolism and insulin action. L6 myotubes possess many important characteristics relevant to our research, including high insulin sensitivity and a similar mitochondrial respiration sensitivity compared to primary muscle fibres. Furthermore, several studies have demonstrated the utility of L6 myotubes as a model for studying insulin sensitivity and metabolism, including our own previous work (PMID: 19805130, 31693893, 19915010) and work of others (PMID:12086937, 29486284, 15193147).

Importantly, our observations from the L6 myotube model are supported by in vivo data from both mice and humans. Chow (Figure 3J, K) and high-fat fed mice (new data - Supplementary Figure 4 H-I) demonstrated a reduction in mitochondrial Ceramide and an increase in CoQ9. Muscle biopsies from humans showed a strong negative correlation between mitochondrial C18:0 ceramide levels and insulin sensitivity (PMID: 29415895). Further, complex I and IV abundance was strongly correlated with both muscle insulin sensitivity and mitochondrial ceramide (CerC18:0) (Figure 6E, F). This is consistent with our observations in L6 myotubes (Figure 6H, I). These findings support the relevance of our in vitro results to in vivo muscle metabolism.

**Points from reviewer 1**
1. Although the authors' results suggest that higher mitochondrial ceramide levels suppress cellular insulin sensitivity, they rely solely on a partial inhibition (i.e., 30%) of insulin-stimulated GLUT4-HA translocation in L6 myocytes. It would be critical to examine how much the increased mitochondrial ceramide would inhibit insulin-induced glucose uptake in myocytes using radiolabeled deoxy-glucose.Another important question to be addressed is whether glycogen synthesis is affected in myocytes under these experimental conditions. Results demonstrating reductions in insulin-stimulated glucose transport and glycogen synthesis in myocytes with dysfunctional mitochondria due to ceramide accumulation would further support the authors' claim.

Response: We have now conducted additional experiments focusing on glycogen synthesis as a readout of insulin sensitivity, as it offers an orthogonal method for assessing GLUT4 translocation and glucose uptake. L6-myotubes overexpressing the mitochondrial-targeted ASAH1 construct (as described in Fig. 3) were challenged with palmitate and insulin stimulated glycogen synthesis was measured using 14C radiolabeled glucose. As shown below, palmitate suppressed insulin-induced glycogen synthesis, which was effectively prevented by overexpression of ASAH1 (N = 5, * p<0.05) supporting our previous observation using GLUT4 translocation as a readout of insulin sensitivity (Fig. 3). These results provide additional evidence highlighting the role of dysfunctional mitochondria in muscle cell glucose metabolism.

These data have now been added to Supplementary Figure 4K and the results modified as follows:

“...For this reason, several in vitro models have been employed involving incubation of insulin sensitive cell types with lipids such as palmitate to mimic lipotoxicity in vivo. In this study we have used cell surface GLUT4-HA abundance as the main readout of insulin response...”

“Notably, mtASAH1 overexpression protected cells from palmitate-induced insulin resistance without affecting basal insulin sensitivity (Fig. 3E). Similar results were observed using insulin-induced glycogen synthesis as an orthologous technique for Glut4 translocation. These results provide additional evidence highlighting the role of dysfunctional mitochondria in muscle cell glucose metabolism (Sup. Fig. 5K). Importantly, mtASAH1 overexpression did not rescue insulin sensitivity in cells depleted…”

Additionally, the following text was added to the method section:

“L6 myotubes overexpressing ASAH were grown and differentiated in 12-well plates, as described in the Cell lines section, and stimulated for 16 h with palmitate-BSA or EtOH-BSA, as detailed in the Induction of insulin resistance section.

On day seven of differentiation, myotubes were serum starved in DMEM for 3.5 h. After incubation for 1 h at 37 °C with 2 µCi/ml D-[U-14C]-glucose in the presence or absence of 100 nM insulin, glycogen synthesis assay was performed, as previously described (Zarini S. et al., JLipid Res, 63(10): 100270, 2022).”

2. In addition, it would be critical to assess whether the increased mitochondrial ceramide and consequent lowering of energy levels affect all exocytic pathways in L6 myoblasts or just the GLUT4 trafficking. Is the secretory pathway also disrupted under these conditions?

Response: This is an interesting point raised by the reviewer that is aimed at the next phase of this work, to identify how ceramide induced mitochondrial dysfunction drives insulin resistance. Looking at energy deficiency in more detail as well as general trafficking is part of ongoing work, but given the complexity of this question, it is beyond the scope of the current study.

**Points from reviewer 2**
1. The mechanistic aspect of the work and conclusions put forth rely heavily on studies performed in cultured myocytes, which are highly glycolytic and generally viewed as a poor model for studying muscle metabolism and insulin action. Nonetheless, the findings provide a strong rationale for moving this line of investigation into mouse gain/loss of function models.

Response: We acknowledge that in vitro models may not fully mimic in vivo complexity as described above in the response to the “Assessment note”. We have now added to theDiscussion:

“In this study, we mainly utilised L6-myotubes, which share many important characteristics with primary muscle fibres. Both types of cells exhibit high sensitivity to insulin and respond similarly to maximal doses of insulin, with GLUT4 translocation stimulated between 2 to 4 times over basal levels in response to 100 nM insulin (as shown in Fig. 1-4 and (46,47)). Additionally, mitochondrial respiration in L6-myotubes has a similar sensitivity to mitochondrial poisons, as observed in primary muscle fibres (as shown in Fig. 5 (48)). Finally, inhibiting ceramide production increases CoQ levels in both L6-myotubes and adult muscle tissue (as shown in Fig. 2-3). Therefore, L6-myotubes possess the necessary metabolic features to investigate the role of mitochondria in insulin resistance, and this relationship is likely applicable to primary muscle fibres”.

2. One caveat of the approach taken is that exposure of cells to palmitate alone is not reflective of in vivo physiology. It would be interesting to know if similar effects on CoQ are observed when cells are exposed to a more physiological mixture of fatty acids that includes a high ratio of palmitate, but better mimics in vivo nutrition.

Response: We appreciate the reviewer's comment. Previously, we reported that mitochondrial CoQ depletion occurs in skeletal muscle after 14 and 42 days of HFHSD feeding, coinciding with the onset of insulin resistance (PMID: 29402381, see Author response image 1).

These data demonstrated that our in vitro model recapitulates the loss of CoQ in insulin resistance observed in muscle tissue in response to a more physiological mixture of fatty acids. Further, it has been reported that different fatty acids can induce insulin resistance via different mechanisms (PMID:20609972), which would complicate interpretation of the data. Saturated fatty acids such as palmitate increase ceramides in cell-lines and humans, but unsaturated FAs generally do not (PMID: 10446195,14592453,34704121). As such we conclude that palmitate is a cleaner model for studying the effects of ceramide on skeletal muscle function.

We have added to discussion:

“…These findings align with our earlier observations demonstrating that mice exposed to HFHSD exhibit mitochondrial CoQ depletion in skeletal muscle (Fazakerley et al. 2018).”

3. While the utility of targeting SMPD5 to the mitochondria is appreciated, the results in Figure 5 suggest that this manoeuvre caused a rather severe form of mitochondrial dysfunction. This could be more representative of toxicity rather than pathophysiology. It would be helpful to know if these same effects are observed with other manipulations that lower CoQ to a similar degree. If not, the discrepancies should be discussed.

Response: As the reviewer suggests many of these lipids can cause cell death (toxicity) if the dose is too high. We have previously found that low levels (0.15 mM) of palmitate were sufficient to trigger insulin resistance without any signs of toxicity (Hoehn, K, PNAS, 19805130). Using a similar approach, we show that mitochondrial membrane potential is maintained in SMPD5 overexpressing cells (Sup. Fig. 2J - and Author response image 2). Given that toxicity is associated with a loss of mitochondrial membrane potential (eg., 50uM Saclac; RH panel), these data suggest SMPD5 overexpression is not causing overt toxicity.

**Author response image 2. sa3fig2:** 

Furthermore, we conducted an overrepresentation analysis of molecular processes within our proteomic data from SMPD5-overexpressing cells. As depicted below, no signs of cell toxicity were observed in our model at the protein level. This data is now available in supplementary table 1.

**Author response image 3 sa3fig3:** 

**Author response image 4. sa3fig4:** 

Our results are therefore consistent with a pathological condition induced by elevated levels of ceramides independently of cellular toxicity. The following text has been added to the discussion:“...downregulation of the respirasome induced by ceramides may lead to CoQ depletion.Despite the significant impact of ceramide on mitochondrial respiration, we did not observe anyindications of cell damage in any of the treatments, suggesting that our models are not explained by toxicity and increased cell death (Sup. Fig. 2H & J).”

4. The conclusions could be strengthened by more extensive studies in mice to assess the interplay between mitochondrial ceramides, CoQ depletion and ETC/mitochondrial dysfunction in the context of a standard diet versus HF diet-induced insulin resistance. Does P053 affect mitochondrial ceramide, ETC protein abundance, mitochondrial function, and muscle insulin sensitivity in the predicted directions?

Response: We agree with the referee about the importance of performing in vivo studies to corroborate our in vitro data. We have now conducted extensive new studies in mice skeletal muscle using targeted metabolomic and lipidomic analyses to investigate the impact of ceramide depletion in CoQ levels in HF-fed mice. Mice were exposed to a HF-fed diet with or without the administration of P053 (selective inhibitor of CerS1) for 5 weeks. As illustrated in Author response image 3, the administration of P053 led to a reduction in ceramide levels (left panel), increase in CoQ levels (middle panel) and a negative correlation between these molecules (right panel), which is consistent with our in vitro findings.

Additional suggestions:1. Figure 1: How does increased mitochondrial ceramide affect fatty acid oxidation (FAO) in L6-myocytes? As the accumulation of mitochondrial ceramide inhibits respirasome and mitochondrial activity in vitro, can reduce FAO in vivo, due to high mitochondrial ceramide, accounts for ectopic lipid deposition in skeletal muscle of obese subjects?

Response: We appreciate the reviewer for bringing up this intriguing point. We would like to emphasise that Complex II activity is vital for fatty acid oxidation. As shown in Fig. 5H, our results indicate that specifically Complex II mediated respiration was diminished in cells with SMPD5 overexpression, suggesting that ceramides hinder the mitochondria's capability to oxidise lipids. We agree that this mechanism may potentially play a role in the ectopic lipid accumulation seen in individuals with obesity.

We have added the following text to discussion:

“...the mitochondria to switch between different energy substrates depending on fuel availability, named “metabolic Inflexibility”...this mechanism may potentially play a role in the ectopic lipid accumulation seen in individuals with obesity, a condition linked with cardio-metabolic disease.”

2. Figure 2: Although the authors show that mtSMPD5 overexpression does not affect ceramide abundance in whole cell lysate, it would be critical to examine the abundance of this lipid in other cellular membranes and organelles, particularly plasma membrane. What is the effect of mtSMPD5 overexpression on plasma membrane lipids composition? Does that affect GLUT4-containing vesicles fusion into the plasma membrane, possibly due to depletion of v-SNARE or tSNARE?Response: While we acknowledge the importance of this point we strongly feel that measuring lipids in purified membranes has its limitations because it is impossible to purify specific membranes without contamination from other kinds of membranes. For example, we have done proteomics on purified plasma membranes from different cell types and we always observe considerable mitochondrial contamination with these membranes (e.g. PMID 21928809). This was the main factor that led us to use the mitochondrial targeting approach.

Nevertheless we do acknowledge that there is a possibility that ceramides that are produced in the mitochondria in SMPD5 cells could leak out of mitochondria into other membranes and this could influence other aspects of GLUT4 trafficking and insulin action. However, we believe that the studies using mito targeted ASAH mitigate against this problem. Thus, we have now included a statement in the revised manuscript as follows: “It is also possible that ceramides generated within mitochondria in SMPD5 cells leak out from the mitochondria into other membranes (e.g. PM and Glut4 vesicles) affecting other aspects of Glut4 trafficking and insulin action. However, the observation that ASAH1 overexpression reversed IR without affecting whole cell ceramides argues against this possibility.”.

3. Figure 4: One critical piece of information missing is the effect (if any) of mitochondrial ceramide accumulation on the mRNAs encoding the ETC components affected by this lipid. Although the ETC protein's lower stability may account for the effect of increased ceramide, transcriptional inhibition can't be ruled out without checking the mRNA expression levels for these ETC components.Response: To address this point, we have quantified the mRNA abundance of nine complex I subunits that exhibit downregulation in our proteomic dataset subsequent to mtSMPD5 overexpression (as depicted in Figure 4G).

Induction of mtSMPD5 expression with doxycycline (Author response image 4 - Left hand panel) had no effect on the mRNA levels of the Complex I subunits (Author response image 4 - right hand panel).. This is consistent with our initial hypothesis that the reduction in electron transport chain (ETC) components, caused by heightened ceramide levels, primarily arises from alterations in protein stability rather than gene expression. While we acknowledge the possibility that certain subunits might be regulated at the transcriptional level, the absence of mRNA downregulation across our data strongly suggests that, at the very least, a portion of the observed protein depletion is attributed to diminished protein stability. We have incorporated this dataset into Supplementary Figure 6J and added the following text to the results:

**Author response image 5. sa3fig5:** 

“Importantly, CI downregulation was not associated with reduction in gene expression as shown in Sup. Fig. 6J.”

Additionally, we have added the following text to discussion:

“In addition, the absence of mRNA downregulation in mtSMPD5 overexpressing cells strongly suggests that at least a portion of the observed protein depletion within CI is attributed to diminished protein stability.”

4. Figure 3: The authors state that neither palmitate nor mtASAH1 overexpression affected insulin-dependent Akt phosphorylation. However, the results in Figure 3F-G do not support this conclusion, as the overexpression of mtASAH1 does enhance the insulin-stimulated AKT (thr-308) phosphorylation. They need to clarify this issue.

Response: We have now analysed these data in a manner that preserves the control variance, consistent with the other figures in the manuscript and there is no significant change in Akt phosphorylation in ASAH over-expressing cells.

**Author response image 6. sa3fig6:** 

5. Figure S2: A functional assessment of mitochondrial function in HeLa cells would be helpful to validate the small effect of Saclac treatment on CI NDUFB8.

Response: Mitochondrial respiration was conducted in cells treated with Saclac (2 µM and 10 µM) for 24 hours. As shown in Author response image 6, in Hela cells, we did not detect any mitochondrial respiratory impairments at low doses, but only at high doses of Saclac. This suggests that the minor effect of Saclac on CI NDUFB8 is insufficient to alter mitochondrial function.

**Author response image 7. sa3fig7:** 

**Reviewer #2 (Recommendations For The Authors):**
Additional questions and comments for consideration:1. The working model links ceramide-induced CoQ depletion to a reduction in ETC proteins and accompanying deficits in OxPhos capacity. The idea that mitochondrial dysfunction necessarily precedes and causes insulin resistance has been heavily debated for years because many animal and human studies have found no overt changes in ETC proteins and/or mitochondrial respiratory capacity during the early phases of insulin resistance. How do the investigators reconcile their work in the context of this controversy?

Response: We acknowledge this controversy in our revised manuscript more clearly now as follows on page 21: “We present evidence that mitochondrial dysfunction precedes insulin resistance. However, previous studies have failed to observe changes in mitochondrial morphology, respiration or ETC components during early stages of insulin resistance (72). However, in many cases such studies fail to document changes in insulin-dependent glucose metabolism in the same tissue as was used for assessment of mitochondrial function. This is crucial because we and others do not observe impaired insulin action in all muscles from high fat fed mice for example. In addition, surrogate measures such as insulin-stimulated Akt phosphorylation may not accurately reflect tissue specific insulin action as demonstrated in figure 1C. Thus, further work is required to clarify some of these inconsistencies''.

2. While the utility of targeting SMPD5 to the mitochondria is appreciated, the results in Figure 5 suggest that this manoeuvre caused a rather severe form of mitochondrial dysfunction. Is this representative of pathophysiology or toxicity?

Response: We believe we have addressed this in point 3 above (Principal comments, reviewer 1, point 3)

3. How did this affect other mitochondrial lipids (e.g. cardiolipin)?

Response: As shown in the supplementary figure 3, SMPD5 overexpression did not affect other lipids species such as cardiolipin (D-J). We have added to results:

“Importantly, mtSMPD5 overexpression did not affect ceramide abundance in the whole cell lysate nor other lipid species inside mitochondria such as cardiolipin, cholesterol and DAGs (Sup. Fig. 3 A, D-J)”

4. Are these severe effects rescued by CoQ supplementation?

Response: We have performed additional experiments to address this point. As shown below, mitochondrial ceramide accumulation induced by palmitate was not reversed by CoQ supplementation, as demonstrated in Figure 1F. We have added to results:

“Addition of CoQ9 had no effect on control cells but overcame insulin resistance in palmitate treated cells (Fig. 1A). Notably, the protective effect of CoQ9 appears to be downstream of ceramide accumulation, as it had no impact on palmitate-induced ceramide accumulation (Fig. 1E-F). Strikingly, both myriocin and CoQ9…”

Additionally, we assessed mitochondrial respiration by using SeaHorse in cells with SMPD5 overexpression treated with or without CoQ supplementation. Our results, depicted below, indicate that CoQ supplementation reversed the ceramide-induced decrease in basal and ATP linked mitochondrial respiration. We have modified Fig.5.

**Author response image 8. sa3fig8:** 

We have added to results:

“Respiration was assessed in intact mtSMPD5-L6 myotubes treated with CoQ9 by Seahorse extracellular flux analysis. mtSMPD5 overexpression decreased basal and ATP-linked mitochondrial respiration (Fig. 5 A, B &C), as well as maximal, proton-leak and non-mitochondrial respiration (Fig. 5 A, D, E & F) suggesting that mitochondrial ceramides induce a generalised attenuation in mitochondrial function. Interestingly, CoQ9 supplementation partially recovered basal and ATP-linked mitochondrial respiration, suggesting that part of the mitochondrial defects are induced by CoQ9 depletion. The attenuation in mitochondrial respiration is consistent with a depletion of the ETC subunits observed in our proteomic dataset (Fig. 4)...”

5. Are these same effects observed with other manipulations that lower CoQ to a similar degree?

Response: As mentioned in point 5 (additional suggestions from Reviewer 1), we conducted mitochondrial respiration measurements on HeLa cells treated with Saclac (2 µM and 10 µM) for 24 hours. Our findings showed no signs of mitochondrial respiratory impairments at low doses of Saclac in HeLa cells, despite observing CoQ depletion at this dose (Fig. Sup. 2C). We believe that this variation could be due to the varying sensitivity of mitochondrial respiration/ETC abundance to ceramide-induced CoQ depletion in different cell lines. Alternatively, it is possible that reduced mitochondrial respiration is a secondary event to other mitochondrial/cellular defects such as mitochondrial fragmentation or deficient nutrient transport inside mitochondria.

6. The mitochondrial concentrations of CoQ required to maintain insulin sensitivity in L6 myocytes seem to vary from experiment to experiment. Is it the absolute concentration that matters and/or the change relative to a baseline condition?

Response: This is an excellent observation. The findings indicate that the absolute concentration of CoQ is the determining factor for insulin sensitivity, rather than the relative depletion of CoQ compared to basal conditions. We have added to discussion:“Finally, mtASAH1 overexpression increased CoQ levels. In both control and mtASAH1 cells, palmitate induced a depletion of CoQ, however the levels in palmitate treated mtASAH1 cells remained similar to control untreated cells (Fig. 3I). This suggests that the absolute concentration of CoQ is crucial for insulin sensitivity, rather than the relative depletion compared to basal conditions, thus supporting the causal role of mitochondrial ceramide accumulation in reducing CoQ levels in insulin resistance”

7. Considering that CoQ has been shown to have antioxidant properties, does the rescue observed after a 16 h treatment require the prolonged exposure, or alternatively, are similar effects observed during short-term exposures (~1-2 h), which might imply a different or additional mechanism.

Response: This is an excellent point that we have long considered. The problem is how to address the question in a way that will be definitive and we are concerned that the experiment suggested by the referee will not generate definitive data. A major issue is that CoQ has low solubility and needs to reach the right compartment. As such if short term treatment (as suggested) does not rescue, it would be difficult to make any definite conclusions as this might just be because insufficient CoQ is delivered to mitochondria. Conversely, if short term treatment does rescue this could be either because CoQ does get into mitochondria and regulate ETC or because of its general antioxidant function. So, even if we observe a rescue after 1 hour of incubation with CoQ, it will not clarify whether this is due to the antioxidant effect or simply because 1 hour is adequate to boost mitoCoQ levels. Thus, in our view this experiment might not get us any closer to the answer. Nevertheless, we do feel this is an important point and we have added the following statement to our revised manuscript to acknowledge this: “Because CoQ can accumulate in various intracellular compartments, it's important to consider that its impact on insulin resistance might be due to its overall antioxidant properties rather than being limited to a mitochondrial effect”

8. In Figure 1, CoQ depletion due to 4NB treatment resulted in increased ceramide levels. Could this be due to impaired palmitate oxidation leading to rerouting of intracellular palmitate to the ceramide pathway? This could be tested using stable isotope tracers.

Response: We have added the statement below to the manuscript to address this point. We feel that while an interesting experiment to perform it is somewhat outside of the major focus of this study.

“One possibility is that CoQ directly controls ceramide turnover (35). An alternate possibility is that CoQ inside mitochondria is necessary for fatty acid oxidation (12) and CoQ depletion triggers lipid overload in the cytoplasm promoting ceramide production (36). Future studies are required to determine how CoQ depletion promotes Cer accumulation. Regardless, these data indicate that ceramide and CoQ have a central role in regulating cellular insulin sensitivity.”

9. To a similar point, it would be helpful to know if the C2 ceramide analog is sufficient to cause elevated mito-ceramide and/or CoQ depletion. If not, the results might imply mitochondrial uptake of palmitate is required.

Response: We feel this point is analogous to Point 7 above in that this experiment is not definitive enough to make any clear conclusions as it may or may not work for many different reasons. For example, C2 ceramide may not work simply because it has the wrong chain length.Moreover, it is clear that C2 ceramide has effects that clearly differ from those observed with palmitate most notably the inhibitory effect on Akt signalling. For these reasons we do not agree with the logic of this experiment.

We have mentioned in the results section:

“Based on these data we surmise that C2-ceramide does not faithfully recapitulate physiological insulin resistance, in contrast to that seen with incubation with palmitate”.

10. Likewise, does inhibition of CPT1 ameliorate or exacerbate palmitate-induced insulin resistance?

Response: This experiment has been performed by a number of different labs. For instance, muscle specific CPT1 overexpression is protective against high fat diet induced insulin resistance in mice (Bruce C, PMID19073774), CPT1 overexpression protects L6E9 muscle cells from fatty acid-induced insulin resistance (Sebastian D, PMID17062841) and increased beta-oxidation in muscle cells enhances insulin stimulated glucose metabolism and is protective against lipid induced insulin resistance (Perdomo G, PMID15105415). We have now cited all of these studies in our revised manuscript in the discussion: “In fact, increased fatty acid oxidation is protective against insulin resistance in several model organisms (37–39)”

11. Does the addition of palmitate to the cells treated with mtSMPD5 further reduce CoQ9 (Figure 2I and 2J)?

Response: This intriguing observation, as highlighted by the referee, has prompted us to conduct additional experiments to investigate the effects of palmitate and SMPD5 overexpression on Coenzyme Q (CoQ) levels in L6 myotubes. As demonstrated in the figures presented below, both palmitate and SMPD5 overexpression independently resulted in the depletion of CoQ9, with no observed additive effects suggesting that they shared a common pathway driving CoQ9 deficiency. One plausible hypothesis is that ceramides may trigger the depletion of a specific CoQ9 pool localised within the inner mitochondrial membrane, likely the pool associated with Complex I (CI) in the Electron Transport Chain (ETC). This hypothesis is supported by previous studies indicating that approximately ~25 - 35 % of CoQ binds to CI (PMID: 33722627) and our data demonstrating that ceramide induces a selective depletion of CI in L6 myotubes (Fig. 4).

We have added this result to Fig. 2I in the main section.

We have added to the result section:

“Mitochondrial CoQ levels were depleted in both palmitate-treated and mtSMPD5-overexpressing cells without any additive effects. This suggests that these strategies to increase ceramides share a common mechanism for inducing CoQ depletion in L6 myotubes (Fig. 2I).”

We have added to the discussion section:

“...These are known to form supercomplexes or respirasomes where ~25 - 35 % of CoQ is localised in mammals (58,16).…The observation that both palmitate and SMPD5 overexpression trigger CoQ depletion without additive effects support the notion that ceramides may trigger the depletion of a specific CoQ9 pool localised within the inner mitochondrial membrane.”

12. Some of the cell-based experiments appear to be underpowered and therefore confidence in the interpretations might benefit from additional repeats. For example, in Figure 3i, it appears that palmitate still causes a substantial reduction of CoQ in the cells treated with mtASAH1, even though mito-ceramide levels are restored to baseline. Please specify if these and other results are representative of multiple cell culture experiments or a single experiment.

Response: Author response image 9 All data were derived from a minimum of 3-4 independent experiments from at least two separate cultures of L6 cells. Separate batches of drug treatments were prepared for each experiment. We have previously compared metabolic parameters between batches of cells differentiated at different times (i.e. at least weeks apart) in a previous study (Krycer PMID 31744882) and found variations of <20% for insulin-stimulated glucose oxidation. With an expected variance of 20% and a type I error rate of 0.05, this is sufficient to detect a 40% difference with a power of 0.8. As the reviewer has indicated this is likely underpowered in situations where variance is unexpectedly high or if a small difference needs to be detected.

**Author response image 9. sa3fig9:** 

In terms of Fig3, the reviewer raises an interesting point. As discussed in point 6, the fact that palmitate still appears to cause a depletion of CoQ in mtASAH1 cells likely indicates that the absolute concentration of CoQ is the determining factor for insulin sensitivity, rather than the relative depletion of CoQ compared to basal conditions. We have added to the discussion:

“Finally, mtASAH1 overexpression increased CoQ levels. In both control and mtASAH1 cells, palmitate induced a depletion of CoQ, but this effect was less pronounced in the mtASAH1 cell line (Fig. 3I). Our results suggest that the absolute concentration of CoQ is crucial for insulin sensitivity, rather than the relative depletion compared to basal conditions, thus supporting the causal role of mitochondrial ceramide accumulation in reducing CoQ levels in insulin resistance”

13. The color scheme of 2E is inconsistent with other panels in the figure.

Response: Corrected

14. It would be helpful if the axis labels for CoQ graphs were labeled as "Mito-CoQ" for clarity.

Response: Corrected